# Polη O-GlcNAcylation governs genome integrity during translesion DNA synthesis

Xiaolu Ma[1], Hongmei Liu[2], Jing Li [3], Yihao Wang[4], Yue-He Ding[5], Hongyan Shen [1], Yeran Yang[1], Chenyi Sun[1], Min Huang[1], Yingfeng Tu[2], Yang Liu[1], Yongliang Zhao[1], Meng-Qiu Dong [5], Ping Xu[4], Tie-Shan Tang[2] & Caixia Guo[1]

DNA polymerase η (Polη) facilitates translesion DNA synthesis (TLS) across ultraviolet (UV) irradiation- and cisplatin-induced DNA lesions implicated in skin carcinogenesis and chemoresistant phenotype formation, respectively. However, whether post-translational modifications of Polη are involved in these processes remains largely unknown. Here, we reported that human Polη undergoes O-GlcNAcylation at threonine 457 by O-GlcNAc transferase upon DNA damage. Abrogation of this modification results in a reduced level of CRL4[CDT2]-dependent Polη polyubiquitination at lysine 462, a delayed p97-dependent removal of Polη from replication forks, and significantly enhanced UV-induced mutagenesis even though Polη focus formation and its efficacy to bypass across cyclobutane pyrimidine dimers after UV irradiation are not affected. Furthermore, the O-GlcNAc-deficient T457A mutation impairs TLS to bypass across cisplatin-induced lesions, causing increased cellular sensitivity to cisplatin. Our findings demonstrate a novel role of Polη O-GlcNAcylation in TLS regulation and genome stability maintenance and establish a new rationale to improve chemotherapeutic treatment.

[1] CAS Key Laboratory of Genomics and Precision Medicine, Beijing Institute of Genomics, University of Chinese Academy of Sciences, Chinese Academy of Sciences, Beijing 100101, China. [2] State Key Laboratory of Membrane Biology, Institute of Zoology, University of Chinese Academy of Sciences, Chinese Academy of Sciences, Beijing 100101, China. [3] Beijing Key Laboratory of DNA Damage Response, College of Life Sciences, Capital Normal University, Beijing 100048, China. [4] State Key Laboratory of Proteomics National Center for Protein Sciences Beijing, Beijing Proteome Research Center, National Engineering Research Center for Protein Drugs, Beijing Institute of Radiation Medicine, Beijing 102206, China. [5] National Institute of Biological Sciences (Beijing), Beijing 102206, China. Xiaolu Ma, Hongmei Liu, and Jing Li contributed equally to this work. Correspondence and requests for materials should be addressed to T.-S.T. (email: tangtsh@ioz.ac.cn) or to C.G. (email: guocx@big.ac.cn)

Translesion DNA synthesis (TLS) is one mode of DNA damage tolerance, which utilizes multiple specialized DNA polymerases to replicate damaged DNA to maintain genome integrity. One such polymerase, polymerase η (Polη), is specifically required for the accurate replicative bypass of cyclobutane pyrimidine dimers (CPDs) in DNA generated by ultraviolet (UV) radiation[1–4]. Mammalian Polη possesses a polymerase catalytic domain in its N-terminus[5,6], a proliferating cell nuclear antigen (PCNA)-interacting region and a ubiquitin-binding zinc finger domain (UBZ) responsible for its interaction with monoubiquitinated PCNA (mUb-PCNA) in its C-terminus[7,8]. Compelling evidence has shown that Polη is recruited to stalled replication forks after UV[7, 9–11] and cisplatin (cis-diamminedichloroplatinum, CDDP) exposure[12,13]. DNA damage-induced Polη focus formation is dependent upon its UBZ domains[7] and the RAD18 protein[14]. The biological significance of Polη in the bypass of UV-induced CPDs is manifested by diseases in mice and humans lacking normal Polη protein[15–17]. Additionally, Polη is capable of replicating across other types of DNA damage in vitro, including CDDP-induced GpG adducts (Pt-GG)[6,18,19]. Consistently, the expression level of Polη is inversely correlated with CDDP treatment efficacy[20,21]. However, since Polη replicates undamaged DNA with a high error rate of $10^{-2}-10^{-3}$, its recruitment and residence at replication forks has to be stringently regulated[6]. So far, it is known that, in addition to protein–protein interactions[4,22–24], protein post-translational modifications (PTMs), such as phosphorylation[13,25] and ubiquitination[8,26,27], fine-tune Polη recruitment and bypass of CPD lesions after UV radiation. Given its reduced affinity for the DNA beyond the CPD, Polη dissociation after TLS has been suggested to be its intrinsic property[28]. Moreover, deubiquitination of mUb-PCNA by USP1[29], USP10[30], or interferon-stimulated gene 15 modification of PCNA has also been suggested to dictate TLS termination[30]. Nevertheless, given that UV-induced mUb-PCNA can persist long after TLS is completed[31], it remains a conundrum how disassembly of Polη happens in the persistence of mUb-PCNA.

Recently, O-linked β-N-acetylglucosamine (O-GlcNAc) to serine and threonine residues of proteins, termed O-GlcNAcylation[32], is emerging as a key regulator of diverse cellular processes, such as signal transduction and proteasomal degradation[33–35]. Analogous to phosphorylation, O-GlcNAcylation is highly dynamic and has extensive crosstalk with other PTM forms[32]. O-GlcNAc cycling is modified by only one O-GlcNAc transferase (OGT) and only one O-GlcNAcase (OGA) in mammals. Its donor substrate, UDP-GlcNAc, exists in high intracellular concentrations. Aberrant O-GlcNAcylation has been linked to a plethora of human diseases, including cancer[36,37]. Recently, OGT has been found to be recruited to sites of DNA damage[38] and several proteins involved in DNA damage response (DDR) have been reported to be O-GlcNAcylated[35,38–40]. O-GlcNAcylation of H2AX is further found to interfere with its phosphorylation[38]. However, much remains a mystery as for the role of O-GlcNAcylation in the DDR.

In this study, we found that OGT interacts with Polη and promotes Polη O-GlcNAcylation at T457. Although T457A mutation does not impair Polη bypass across CPD lesions, it unexpectedly restrains p97-dependent Polη removal from replication forks after TLS is completed, leading to an increased mutation frequency after UV irradiation. Polη also interacts with DDB1 and CDT2. Intriguingly, T457A mutation significantly attenuates Lys48 (K48)-linked Polη polyubiquitination catalyzed by Cullin 4-RING Ligase (CRL4)-DDB1-CDT2 (CRL4$^{CDT2}$), explaining the prolonged retention of the Polη T457A mutant at UV-damaged chromatin. Through quantitative mass spectrometry (MS), we found that the Polη T457A mutant displays an obvious reduction in ubiquitination at K462, revealing a novel

crosstalk between Polη O-GlcNAcylation and ubiquitination, which modulates Polη disassembly from replication forks. Additionally, T457A mutation also causes a TLS deficiency upon CDDP exposure, accompanied with a reduced DNA replication rate. Therefore, O-GlcNAcylation plays an unexpected role in TLS polymerase switching, adding a further layer of regulation that elaborately controls TLS and genome stability in vivo.

## Results

**OGT binds to Polη and promotes Polη O-GlcNAcylation at T457.** To identify novel proteins that may regulate Polη functions in vivo, we transfected HEK293T cells with a 2xFlag-Polη expression vector and performed immunopurification using the nuclear extracts[41]. Affinity-purified proteins associated with Flag-Polη were separated by sodium dodecyl sulfate-polyacrylamide gel electrophoresis (SDS–PAGE) and revealed by silver staining (Fig. 1a). Two indicated regions enriched with bands not observed in the affinity-purified control extracts were cut and several proteins including OGT were identified via Liquid chromatography-tandem MS (LC-MS/MS) analysis. We then performed co-immunoprecipitation (Co-IP) assay to validate OGT/Polη interaction. Endogenous OGT was found to bind to Flag-Polη but not Flag tag alone, although the immunoprecipitated OGT only represents a small fraction (<1%) of that in cells (Fig. 1b). OGT has been shown to relocate to the sites of DNA damage after ionizing radiation treatment. We then checked whether OGT is involved in UV-induced DDR. As shown in Fig. 1c, we found that the chromatin binding of OGT was enhanced after UV treatment. Interestingly, knockdown of OGT did not affect UV-induced Polη focus formation at 8 h but causing a significantly higher extent of Polη foci at 24 h post-UV compared with negative control (siNC) (Fig. 1d), suggesting that OGT might regulate Polη function after UV irradiation.

Given that OGT can add O-GlcNAc to target proteins, exogenously expressed Flag-Polη in 293-T cells was immunoprecipitated under a denaturing condition to assess its O-GlcNAcylated level through immunoblotting using anti-O-GlcNAc antibody. A band corresponding to O-GlcNAcylated Polη was observed (Fig. 1e). After treating the cells with glucose and Thiamet-G, the OGA inhibitor that suppresses the reversible removal of O-GlcNAc moiety from proteins, we found that Polη O-GlcNAcylation was substantially enriched (Fig. 1e). Additionally, 293T cells transfected with green fluorescent protein (GFP)-Polη were cultured in media supplemented with different concentrations of glucose followed by IP under an undenaturing condition. We found that Polη O-GlcNAcylation level positively correlates with the glucose concentration in the medium (Fig. 1f), consistence with the fact that glucose is an important source of UDP-GlcNAc, the high-energy donor substrate of OGT. To exclude the possibility that Polη O-GlcNAcylation might be caused by its overexpression, we took advantage of an inducible XP30RO-Polη cell line[42] to express SFB (streptavidin-Flag–S protein)-tagged Polη close to the endogenous level through supplementing Doxycycline (0.01 µg ml$^{-1}$) in the medium. The result indicated that Polη when expressed at the endogenous level was also O-GlcNAcylated (Supplementary Fig. 1a).

To identify the potential O-GlcNAcylated residue(s) in Polη, 293T cells were transfected with Flag-Polη, and O-GlcNAcylation was enriched by treating the cells with Thiamet-G and glucose[43]. The immunoprecipitated Flag-Polη were divided into two groups: one was separated on SDS-PAGE and the expected Polη band was cut out for in-gel digestion and higher-energy collision dissociation tandem MS (HCD-MS) analysis, the other one was glycine eluted and subjected to HCD-MS analysis directly (Supplementary Fig. 1b, c). Results from both groups (with 92.8% or 86.4% Polη protein

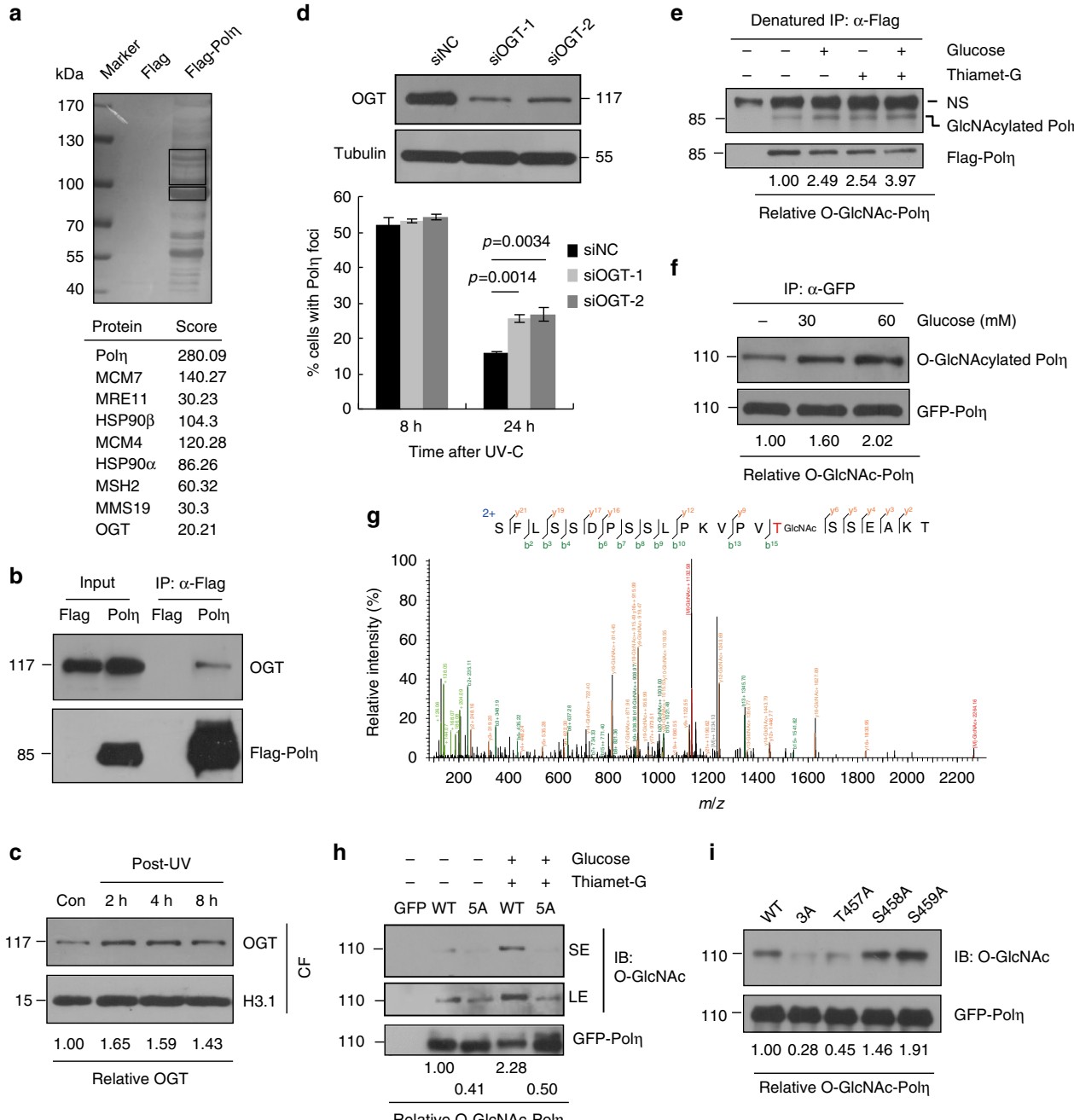

**Fig. 1** Polη interacts with OGT and is subject to O-GlcNAcylation predominantly at T457. **a** The nuclear extracts of 293T cells expressing Flag-Polη were immunoprecipitated. The indicated bands (black rectangles) were cut and analyzed via mass spectrometric analysis. **b** 293T cell lysates expressing Flag-Polη were immunoprecipitated and detected with anti-OGT and anti-Flag antibodies. The input included 2% of the cell lysate used. **c** U2OS cells were irradiated with UVC (15 J m$^{-2}$) and harvested at different time points later. The chromatin fractions were extracted followed by immunoblotting with OGT and H3.1 antibodies. **d** U2OS cells were transfected with siOGT or siNC oligos. The OGT protein level was detected by western blot. The knockdown cells were transfected with GFP-Polη, irradiated with UVC (15 J m$^{-2}$) and further incubated for 24 h. The proportions of GFP-Polη-expressing cells with >30 foci were determined. Data represent means ± SEM from three independent experiments. **e** 293T cells transfected with Flag-Polη or empty vector were incubated with Thiamet-G and glucose. The cell lysates were denatured and immunoprecipitated with anti-Flag M2 beads followed by immunoblotting with O-GlcNAc and Flag antibodies. NS: non-specific band. **f** 293T cells transfected with GFP-Polη were treated with Thiamet-G and different concentrations of glucose. The cell lysates were immunoprecipitated with GFP-Trap A followed by immunoblotting with O-GlcNAc and GFP antibodies. **g** 293T cells transfected with Flag-Polη or empty vector were treated with Thiamet-G and glucose. The cell lysates were immunoprecipitated with anti-Flag agarose followed by HCD-MS analysis. The tryptic peptide containing a HexNAc (+203.08 Da) was detected. The b- and y-type product ions were marked on the spectrum and also illustrated along the peptide sequence shown on top of the spectrum. **h, i** 293T cells transfected with the indicated GFP-Polη constructs were treated as in **g**. The cell lysates were immunoprecipitated with GFP-Trap A followed by immunoblotting with O-GlcNAc and GFP antibodies. SE short exposure, LE long exposure

sequence coverage, respectively) revealed that Polη O-GlcNAcylation mainly occurs on a peptide (SFLSSDPSSLPKVPVTSSEAKTQGSGPAVT) spanning residues 442–471 (Fig. 1g). To further determine the potential modification site(s), we generated a mutated Polη construct (Polη-5A), in which three serine (S) and two threonine (T) residues within the O-GlcNAc-modified peptide were mutated to alanines (A) (T457A/S458A/S459A/S466A/T471A). Wild-type (WT) and Polη-5A constructs were then transfected into 293T cells and their O-GlcNAcylation levels were compared. As shown in Fig. 1h, the O-GlcNAcylation level of Polη-5A was significantly reduced compared to that of WT in the absence or presence of Thiamet-G and glucose, supporting the notion that the majority of Polη O-GlcNAcylation occur within these residues.

Interestingly, both of S466A and T471A single mutants exhibited an increased Polη O-GlcNAcylation, hinting that these two residues were not the major sites for this modification (Supplementary Fig. 1d). We then generated another mutated Polη construct (Polη-3A) (T457A/S458A/S459A) and found that Polη-3A also manifested a markedly reduced level of O-GlcNAcylation as Polη-5A (Supplementary Fig. 1e). Furthermore, we generated three single mutants (T457A, S458A, and S459A) and assessed their O-GlcNAcylation levels. Intriguingly, T457A mutant, but not S458A or S459A mutant, showed a dramatically diminished level of Polη O-GlcNAcylation (Fig. 1i). Thus it is highly likely that T457 is the major O-GlcNAcylation site of Polη. T457 is located at the PVT/S (proline–valine–threonine/serine) motif that has been reported to be conserved in about half of the O-GlcNAcylated proteins identified to date[32]. Additionally, we found that T457A, 3A, and 5A mutants manifest reduced associations with OGT (Supplementary Fig. 1f). Furthermore, we mapped the regions within Polη responsible for its interaction with OGT by using Flag-tagged WT Polη and a series of Polη deletion mutants (Supplementary Fig. 2a). Co-IP experiments revealed that the N-terminal fragment spanning the whole catalytic domain of Polη is required for its interaction with OGT (Supplementary Fig. 2b). Similarly, we also mapped the regions within OGT required for its association with Polη by using Myc-tagged WT OGT and a series of OGT deletion mutants (Supplementary Fig. 2c). Co-IP assays showed that the first two TPR regions of OGT are required for its association with Polη (Supplementary Fig. 2d).

**T457A has no effect on Polη bypass CPDs**. To examine whether O-GlcNAc modification of Polη participates in DDR, 293T cells expressing GFP-Polη were treated with UVC followed by an IP with anti-GFP antibody. We found that O-GlcNAcylation of Polη was dramatically increased after UV irradiation (Fig. 2a, top panel), which also exhibited a dynamic change after UV (Fig. 2a, bottom panel). Moreover, interaction between Polη and OGT was substantially elevated in 293T cells transfected with Flag-Polη under UV radiation (Fig. 2b). These data suggest that O-GlcNAc modification of Polη plays a potential role in cellular response to UV exposure. To investigate the functional importance of Polη O-GlcNAcylation upon UV irradiation, we transfected WT and T457A-mutated GFP-Polη into U2OS cells and compared their focus formation after UV treatment. No significant difference in their focus formation between WT and T457A Polη was detected at 8 h after UV irradiation (Fig. 2c), suggesting that abolishment of Polη O-GlcNAcylation by T457A does not affect UV-induced Polη focus formation. Consistently, Polη-3A and Polη-5A also failed to show any obvious changes in focus formation post-UV irradiation relative to WT Polη (Fig. 2c).

To determine whether T457A mutation affects the ability of Polη to bypass CPD lesions, we established WT or T457A GFP-Polη-complemented XP30RO (Polη-deficient) stable cell lines (Fig. 2d) and treated the cells with UVC (5 J m$^{-2}$). As shown in Fig. 2e, although unreplicated CPDs persisted in the GFP-expressing XP30RO cells, they were undetectable in the XP30RO cells expressing either WT or T457A Polη at 4 h post-UV. It has been reported that TLS is involved in the disappearance of RPA foci[44], which reflects the ssDNA regions in the chromatin. RPA signals were further examined in the above treated cells. Compared to GFP-expressing control cells that had a relative higher percentage of RPA foci-positive cells at either 4 or 24 h post-UV irradiation, WT or T457A Polη-expressing cells only manifested scanty RPA signals at those time points (Fig. 2f, g). These data suggested that T457A mutation does not compromise Polη ability to bypass CPD lesions.

**T457A impairs Polη removal from chromatin after UV**. Although loss of Polη O-GlcNAcylation caused by T457A does not impair Polη focus formation at 8 h after UV irradiation, we observed that cells expressing Polη-T457A, Polη-3A, or Polη-5A displayed a significantly higher extent of Polη foci at 24 h post-UV compared with that of WT control (Fig. 3a). In line with this, our chromatin fractionation result revealed that T457A mutation remarkably impaired the removal of Polη from chromatin at 24 h post-UV (Supplementary Fig. 3a). FACS analysis showed that the cells expressing WT or T457A Polη had comparable cell cycle profiles at 24 h post-UV (Supplementary Fig. 3b), excluding the possibility that the increased Polη foci at that time is the result of a higher proportion of cells in S phase. O-GlcNAcylation often competes with phosphorylation in many biological processes, and to exclude the possibility that the phenotype may result from the loss of phosphorylation at T457, we constructed two phospho-mimetic Polη mutants by replacing Thr457 with Asp (T457D) or Glu (T457E). We found that analogous to T457A mutant, T457D and T457E Polη still displayed a significantly higher extent of Polη foci at 24 h after UV (Supplementary Fig. 3c). We then speculated that the higher extent of Polη foci at 24 h post-UV might be due to their aberrant disassociation from replication forks after TLS is completed. Recently, several studies have reported that Polη disassociation from chromatin is modulated by an AAA-ATPase p97-UFD1-NPL4 complex, which acts as a K48 ubiquitin-chain-dependent protein segregase to remove poly-ubiquitinated DNA-binding proteins at the sites of DNA damage[45–47]. We found that 5A, T457A, and WT Polη manifested similar associations with either UFD1 (Fig. 3b and Supplementary Fig. 4a) or NPL4 (Fig. 3c and Supplementary Fig. 4b), the two core adaptor factors for p97 to process its K48-polyubiquitinated substrates[47]. Interestingly, the polyubiquitination level of Polη was significantly reduced when WT Polη was mutated to T457A or 3A or 5A (Fig. 3d). Using ubiquitin chain-specific K48 antibody, we further confirmed that ablation of Polη O-GlcNAcylation at T457 significantly decreased its K48-polyubiquitination (Fig. 3d), which might restrain the removal of Polη from replication forks after TLS is completed.

Since Polη replicates undamaged DNA with very low fidelity[6,48], we speculated that failure of Polη disassembly from replication forks in time might cause excessive TLS and increased mutation frequencies. To test that, a mutagenesis assay based on a UV–irradiated shuttle vector pSP189 that carries a mutant supF-suppressor tRNA33 was performed[49]. We found that expressing WT but not T457A Polη in XP30RO cells caused a significant reduction in the mutation frequency after exposure to UVC (400 J m$^{-2}$) (Fig. 3e). We also analyzed whether T457A mutation would affect cellular resistance to UV irradiation. Unlike WT Polη that can completely rescue the UV hypersensitivity in XP30RO cells, T457A mutant only displayed a partial rescue

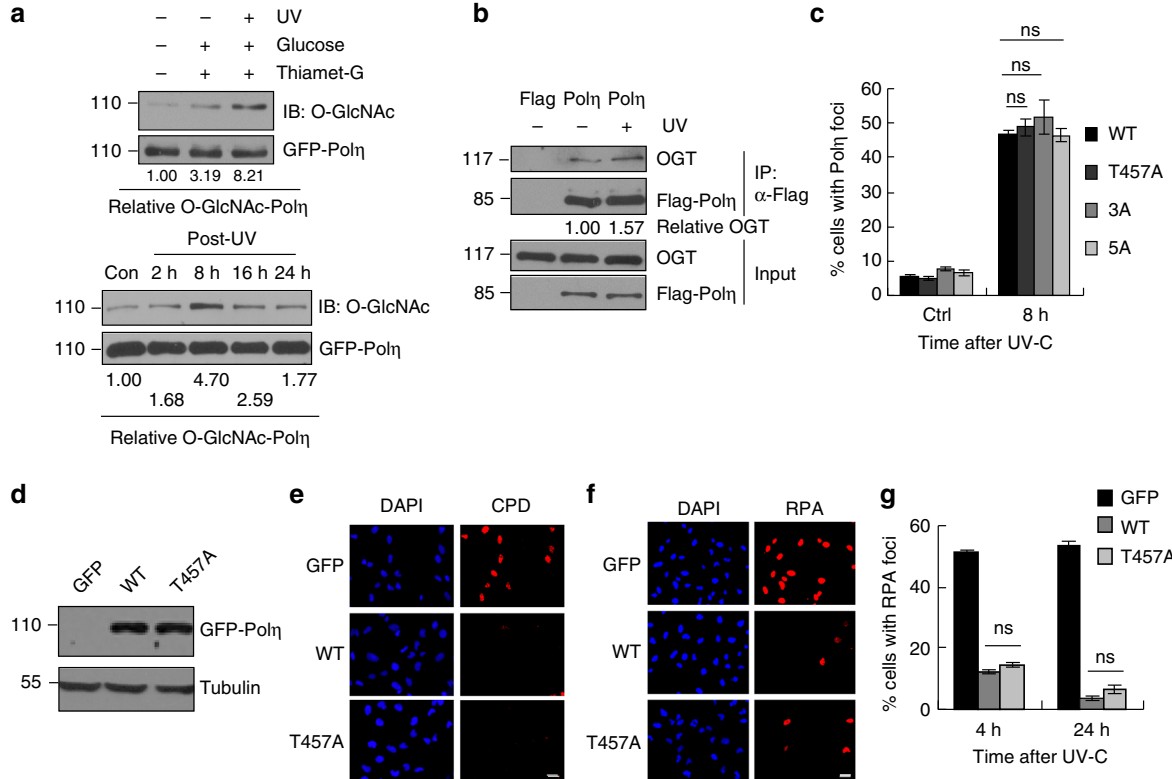

**Fig. 2** T457A has no effect on the efficiency of Polη to bypass CPDs. **a** 293T cells transfected with GFP-Polη were treated with Thiamet-G and glucose. Cells were irradiated with UVC (15 J m$^{-2}$) and harvested 4 h later (top panel) or at the indicated time points (bottom panel). The cell lysates were immunoprecipitated with GFP-Trap A followed by immunoblotting with O-GlcNAc and GFP antibodies. Experiment was repeated three times. SE short exposure, LE long exposure. **b** 293T cells transfected with Flag-Polη were irradiated with UVC (15 J m$^{-2}$) and harvested 4 h later. The cell lysates were immunoprecipitated and the relative IPed OGT was shown. Experiment was repeated twice. **c** U2OS cells transfected with indicated GFP-Polη constructs were UVC irradiated and incubated for 8 h. The proportions of GFP-Polη expressing cells with >30 foci were determined by counting at least 200 cells in each experiment. **d** Polη-deficient XP30RO cells infected with GFP, WT or T457A GFP-Polη lentivirus were selected by flow cytometry. The levels of Polη protein in stable clones were examined with anti-Polη. Tubulin: loading control. **e** Representative images of cells stained with CPDs (red) in ssDNA of nuclei (DAPI, blue) after UV irradiation. XP30RO cells stably expressing GFP, WT or T457A GFP-Polη were irradiated with 5 J m$^{-2}$ UVC and cultured for 4 h. The cells were treated with 1% Triton-X100 for 2 min prior to fixation. Scale bars: 10 μm. **f** Representative images of cells stained with DAPI and RPA2 after UV irradiation. XP30RO cells stably expressing GFP, WT or T457A GFP-Polη were irradiated with 5 J m$^{-2}$ UVC and further cultured. At the indicated time points, cells were treated with 0.5% Triton-X100 for 30 min followed by immunofluorescence. Scale bars: 10 μm. **g** Quantification of the percentage of cells with >10 RPA foci. For each cell line at each time point, at least 250 cells were counted. Data represent means ± SEM from three independent experiments. ns not significant

effect (Fig. 3f). Hence, Polη T457A mutation impairs cellular response to UV radiation. We also determined whether depletion of OGT would affect cellular response to UV radiation. We found that knockdown of OGT in 293T cells significantly abrogated Polη polyubiquitination (Supplementary Fig. 5a) and promoted UV light–induced mutagenesis (Supplementary Fig. 5b). Additionally, depletion of OGT led to an increased UV sensitivity in both U2OS and MRC5 cells (Supplementary Fig. 5c, d). These data support the notion that Polη O-GlcNAcylation at T457 is required for optimal cellular response to UV radiation.

**T457A impairs Polη O-GlcNAcylation and ubiquitination.** We next determined how T457A mutation could remarkably impair Polη polyubiquitination and promote its retention at replication forks. Based on the fact that Polη in *Caenorhabditis elegans* is subjected to CRL4$^{CDT2}$-dependent proteolysis in the context of chromatin-bound PCNA after DNA damage treatment[50] and that the CRL4$^{CDT2}$ E3 ligase can induce ubiquitination of multiple PIP degron-containing proteins[51], we speculated that CRL4$^{CDT2}$ might modulate human Polη polyubiquitination at replication forks and its subsequent degradation. In support of this, we found

that depletion of the CRL4$^{CDT2}$ adaptor protein DDB1 in 293T cells led to an enhanced level of endogenous Polη (Fig. 4a) but a significant reduction in Polη polyubiquitination (Fig. 4b). To further explore whether Polη O-GlcNAcylation at T457 affects this process, both WT and T457A Polη protein levels were tested after silencing the components of CRL4$^{CDT2}$. As shown in Supplementary Fig. 6a, upon efficient reduction of either DDB1 or Cul4A, the level of WT but not T457A Polη was upregulated, suggesting that CRL4$^{CDT2}$ targets WT but not T457A Polη for polyubiquitination and subsequent proteolysis. In support, depletion of either DDB1 or CDT2, the components of CRL4$^{CDT2}$, remarkably decreased the polyubiquitination levels of WT but not T457A Polη in 293T cells (Fig. 4c, d). Additionally, a similar result was obtained when the chromatin-bound WT or T457A Polη were immunoprecipitated (Fig. 4e). To avoid non-specific artifacts caused by aberrant p97-mediated removal of Polη from replication forks, p97 expression was knocked down and the polyubiquitination levels of chromatin bound WT and T457A Polη were compared. p97 depletion failed to change the trend of a significantly higher polyubiquitination level of WT Polη than that of T457A (Fig. 4f). Notably, p97 knockdown increased the polyubiquitination level of WT but not of T457A

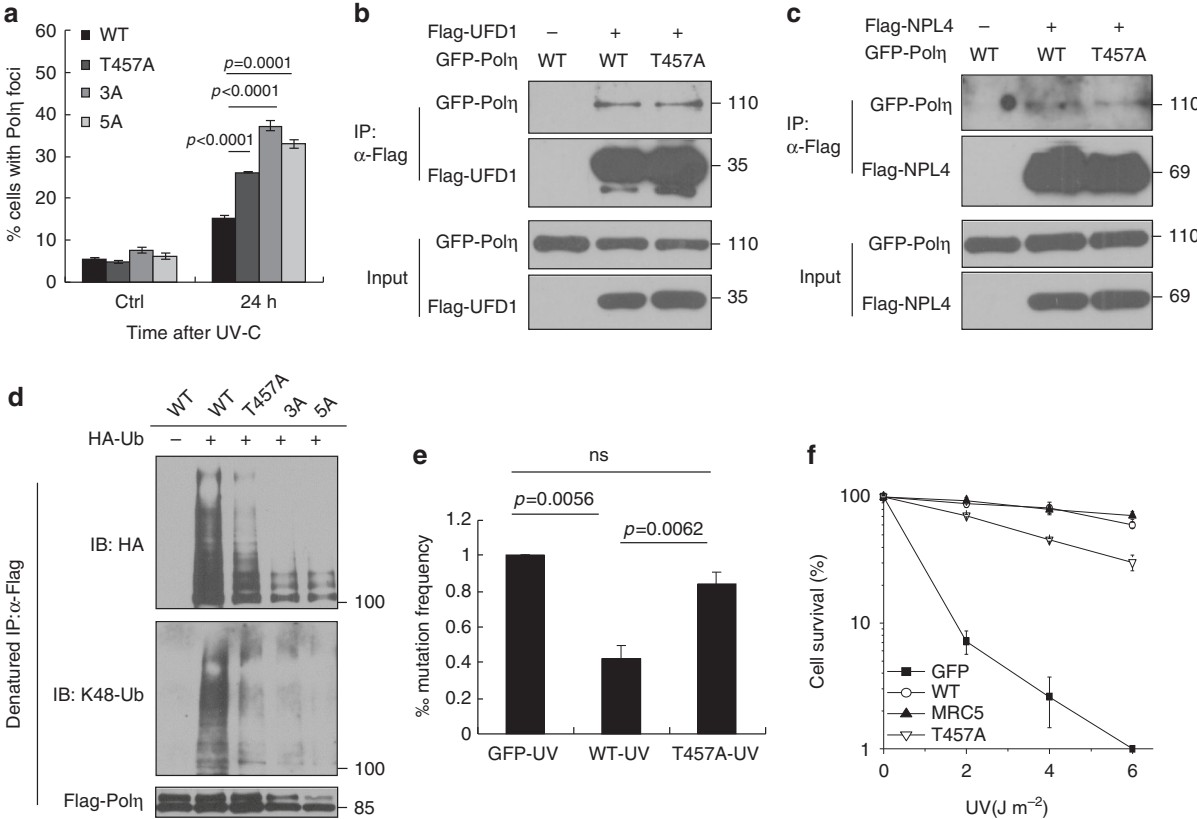

**Fig. 3** T457A impairs Polη removal from chromatin after UV irradiation. **a** U2OS cells transfected with WT or mutated GFP-Polη constructs were irradiated with UVC (15 J m$^{-2}$) and further incubated for 24 h. The proportions of GFP-Polη-expressing cells with >30 foci were determined. Data represent means $\pm$ SEM from three independent experiments. **b**, **c** Flag-UFD1 **b** or Flag-NPL4 **c** and GFP-Polη (WT or T457A) were transfected into 293T cells. The lysates were immunoprecipitated using anti-Flag M2 beads. The inputs and immunoprecipitates were examined via western blotting using anti-GFP and anti-Flag antibodies. **d** 293T cells were transfected with WT or mutant Flag-Polη as well as HA-Ub or HA empty vector. The cell lysates were immunoprecipitated with anti-Flag M2 beads under denaturing condition, followed by immunoblotting with anti-HA, anti-K48-Ubiquitin, and anti-Flag antibodies. **e** Mutation frequency in damaged (400 J m$^{-2}$ UVC) supF plasmid was determined. Data represent means $\pm$ SEM from three independent experiments. ns not significant. **f** XP30RO cells stably expressing GFP, WT or T457A GFP-Polη and MRC5 cells were irradiated with the indicated doses of UVC and further incubated in medium supplemented with 0.4 mM caffeine for 7–10 days. The number of cell clones was determined. Surviving fraction was expressed as a percentage of mock-treated cells. Experiment was repeated three times, giving similar results. The representative curve is shown. Error bar: s.d., $n=3$

Polη (Fig. 4f). To support that O-GlcNAcylation at T457 promotes CRL4$^{CDT2}$-mediated Polη polyubiquitination, we reconstituted the ubiquitination reaction in vitro. His-tagged WT or T457A Polη were co-expressed with GST-OGT in *Escherichia coli* cells, in which WT but not T457A Polη exhibited a strong O-GlcNAcylation signal (Supplementary Fig. 6b, c). The purified WT or T457A His-Polη were then used as substrates for in vitro ubiquitination. Components of CRL4$^{CDT2}$ E3 ligase complex were co-expressed and immunopurified by anti-FLAG antibody and eluted with the FLAG epitope peptide (Supplementary Fig. 6d). The in vitro ubiquitination result showed that, without co-expression of OGT, WT and T457A Polη exhibited similar extent of ubiquitination (Supplementary Fig. 6e). After co-expression of OGT, the ubiquitination level of WT but not of T457A manifested a remarkably increase. These data suggested that Polη O-GlcNAcylation at T457 promotes CRL4$^{CDT2}$-mediated Polη polyubiquitination, which is the prerequisite for p97-modulated Polη removal from chromatin. Consistently, CDT2 depletion significantly delayed the dissociation of WT but not of T457A Polη from replication forks at 24 h after UV irradiation (Supplementary Fig. 6f). To further determine why CRL4$^{CDT2}$ preferentially polyubiquitinates WT instead of T457A Polη, we performed Co-IP experiments and found that T457A mutation significantly decreased the interactions between Polη and DDB1

or CDT2 (Fig. 4g, h), whereas it had no obvious effects on its binding with MDM2, another E3 ligase involved in Polη polyubiquitination[26] (Supplementary Fig. 6g). Therefore, the diminished polyubiquitination of T457A Polη is likely due to its reduced association with the E3 ligase CRL4$^{CDT2}$. We further performed in vitro pull down assay and found that, under the presence of GST-OGT, which promoted Polη O-GlcNAcylation, the association between Polη and DDB1 or CDT2 was significantly enhanced, suggesting that Polη O-GlcNAcylation promotes its interaction with CRL4$^{CDT2}$ complex (Supplementary Fig. 6h). Taken together, these results demonstrated that T457A mutation affects Polη O-GlcNAcylation and Polη association with CRL4$^{CDT2}$ and impairs CRL4$^{CDT2}$-mediated Polη polyubiquitination on chromatin, thereby limiting p97-dependent removal of Polη from replication forks upon completion of TLS.

**T457A impairs Polη polyubiquitination at K462**. Next, we attempted to identify the specific lysine residue(s) in Polη, whose polyubiquitination is not only required for timely removal of Polη from replication forks but also can be abrogated by T457A mutation. We first analyzed the ubiquitination sites in WT and T457A Polη using quantitative MS. As shown in Supplementary Fig. 7a, b, T457A mutation markedly reduced both overall and

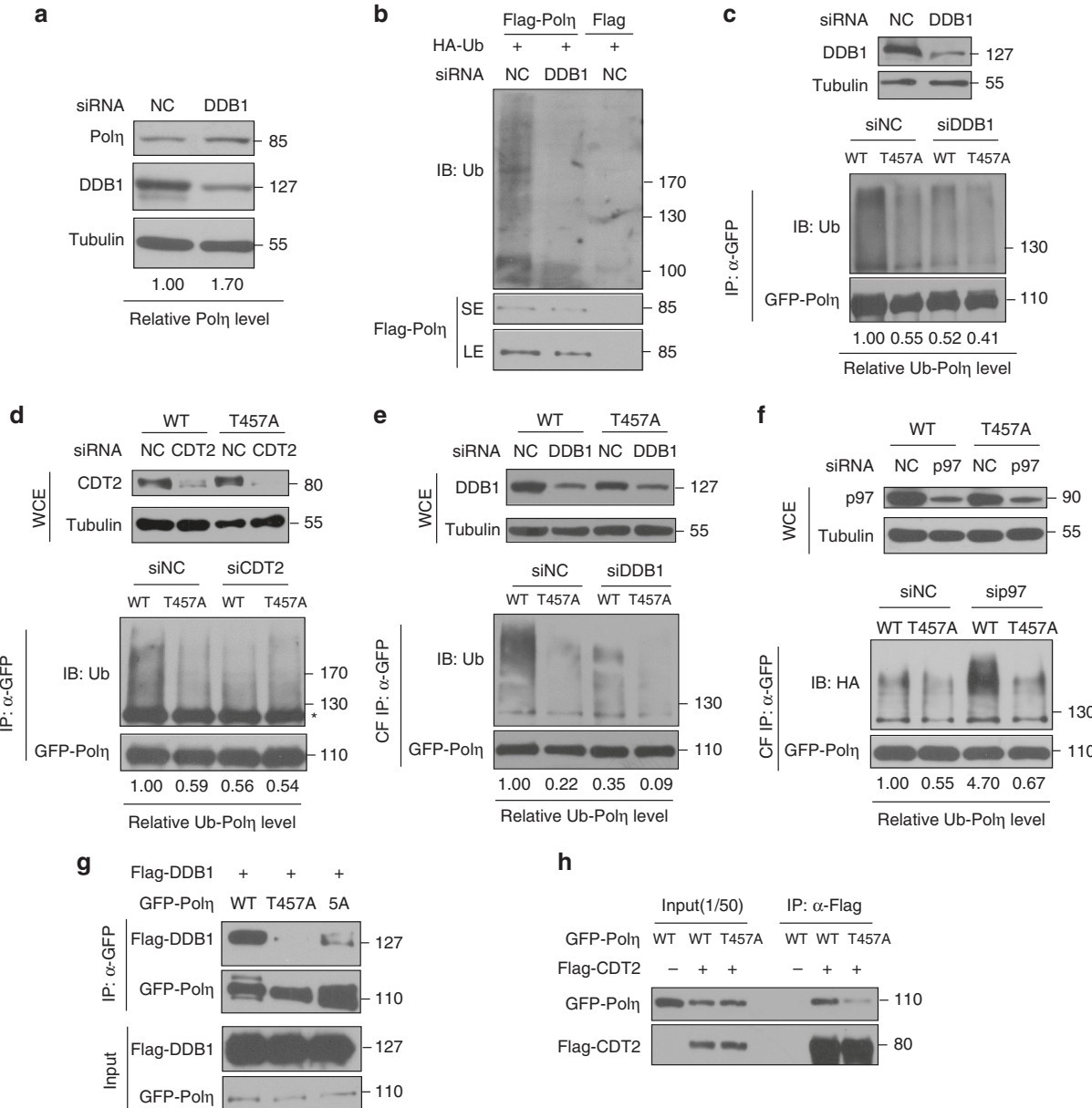

**Fig. 4** T457A impairs CRL4^CDT2-mediated Polη polyubiqitination. **a** 293T cells were transfected with siDDB1 or siNC oligos. Seventy-two hours later, the cells were harvested and the protein levels of Polη and DDB1 were detected by western blotting. Tubulin: loading control. **b** Flag empty vector or Flag-Polη and HA-Ub were transfected in siNC- or siDDB1-treated 293T cells. The cells lysates were immunoprecipitated by anti-Flag M2 beads and analyzed by immunoblotting with anti-Ubiquitin and anti-Flag antibodies. SE short exposure, LE long exposure. **c, d** WT or T457A GFP-Polη and HA-Ub were transfected into siDDB1-treated **c** or siCDT2-treated **d** 293T cells. The lysates were immunoprecipitated using GFP-Trap A. The immunoprecipitates were analyzed via western blotting using anti-GFP and anti-Ubiquitin antibodies. The protein levels of DDB1 and CDT2 were detected by immunoblotting. WCE whole cell lysate. Tubulin: loading control. Asterisk denote non-specific signal. **e** WT or T457A GFP-Polη and HA-Ub were transfected into siDDB1-treated 293T cells. The chromatin fractions (CF) were harvested followed by immunoprecipitation with GFP-Trap A. The immunoprecipitates were analyzed as in **c**. **f** WT or T457A GFP-Polη and HA-Ub were transfected into p97 siRNA (sip97)-treated 293T cells. The chromatin fractions were immunoprecipitated and analyzed as in **c**. The p97 protein level was detected by western blot. **g** Flag-DDB1 and GFP-Polη (WT, T457A, or 5A) were transfected into 293T cells. The lysates were immunoprecipitated using GFP-Trap A. The immunoprecipitates and inputs were examined via western blotting using anti-GFP and anti-Flag antibodies. **h** Flag empty vector or Flag-CDT2 and GFP-Polη (WT or T457A) were transfected into 293T cells. The lysates were immunoprecipitated and analyzed as in **g**

K48-linked ubiquitination levels of Polη, consistent with the result in Fig. 3d. Moreover, seven individual lysine residues (K131, K163, K311, K453, K462, K494, and K682) in the Polη protein were faithfully found to be ubiquitinated (Fig. 5a). Specifically, T457A mutation caused a significant decrease in ubiquitination level at residues K131, K453, and K494 and an even more pronounced reduction at K462 (Fig. 5a and

Supplementary Fig. 7c). Given that protein O-GlcNAc modification usually interplays with other PTMs at adjacent sites, we then focused on K453 and K462 in the vicinity of T457. We mutated K453 and K462 residues to arginine, respectively, to examine their effects on Polη removal after UV radiation. Interestingly, we found that K462R but not K453R mutation compromised the timely Polη disassembly, just analogous to

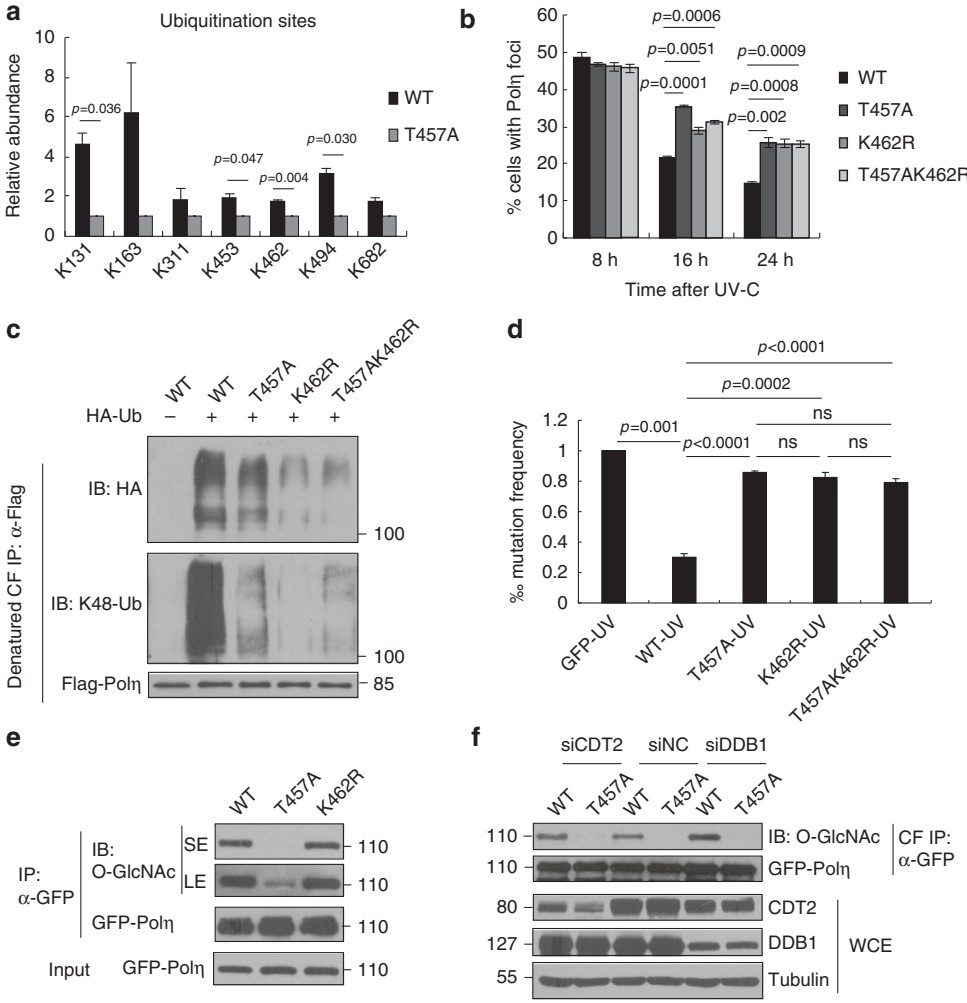

**Fig. 5** T457A impairs Polη polyubiqitination at K462. **a** 293T cells transfected with Flag empty vector, WT, or T457A Flag-Polη and HA-Ub were immunoprecipitated with anti-Flag agarose in a denatured condition followed by quantitative MS analysis as described in "Methods" section. All ubiquitination sites identified in WT and T457A Polη. Error bars: s.e.m. **b** U2OS cells transfected with WT or mutated (T457A, K462R, and T457AK462R) GFP-Polη were irradiated by UVC (15 J m$^{-2}$) and further incubated for 24 h. The proportion of GFP-Polη-expressing cells with >30 foci was determined. Data represent means ± SEM from three independent experiments. **c** 293 T cells were transfected with WT or mutated (T457A, K462R, and T457A/K462R) Flag-Polη and HA-Ub. The collected chromatin fractions (CFs) were denatured followed by immunoprecipitation with anti-Flag agarose. The immunoprecipitates were immunoblotted with anti-HA, anti-K48-Ubiquitin, and anti-Flag antibodies. **d** Mutation frequency in damaged (400 J m$^{-2}$ UVC) *supF* plasmid was determined. Data represent means ± SEM from three independent experiments. ns not significant. **e** WT, T457A, or K462R GFP-Polη were transfected into 293T cells. The cell lysates were immunoprecipitated with GFP-Trap A, followed by immunoblotting with anti-O-GlcNAc and anti-GFP antibodies. "SE" and "LE" represent short and long exposure, respectively. **f** WT or T457A GFP-Polη were transfected into siDDB1-, siCDT2-, or siNC-treated 293T cells. The chromatin fractions were immunoprecipitated with GFP-Trap A and analyzed via western blotting with anti-O-GlcNAc and anti-GFP antibodies

T457A mutation (Fig. 5b and Supplementary Fig. 7d). Meanwhile, T457A/K462R double mutation did not cause a further delay in Polη dissociation from replication forks at 16 and 24 h after UV irradiation compared to either T457A or K462R single mutation (Fig. 5b), suggesting that K462 ubiquitination is downstream of T457 O-GlcNAcylation, one pathway critical in timely removal of Polη from replication forks.

K462 has been reported to be a potential CRL modification site identified in a systematic proteomic study[52], which prompted us to assess whether K462 is a specific ubiquitination site catalyzed by the CRL4$^{CDT2}$ complex. After co-transfecting Polη with Flag-tagged CRL4$^{CDT2}$ adaptor protein DDB1, the polyubiquitination level of Polη was found to be significantly reduced when WT Polη was mutated to K462R (Supplementary Fig. 7e). Then we transfected HA-Ub with WT, T457A, K462R, or T457A/K462R Flag-Polη constructs into 293T cells and

collected the chromatin fractions for denatured IP. We observed that, similar as T457A, K462R mutation also significantly reduced the polyubiquitination of chromatin Polη (Fig. 5c). Using ubiquitin chain-specific K48 antibodies, we further confirmed that K462R mutation remarkably decreased Polη K48-polyubiquitination (Fig. 5c). Consistently, T457A/K462R double mutation did not induce further reduction in Polη K48-polyubiquitination, supporting that K462 polyubiquitination probably sequentially occurs after T457 O-GlcNAc modification. Furthermore, K462R or T457A/K462R Polη-complemented XP30RO cells exhibited similar levels of UV-induced mutation frequency and UV hypersensitivity as T457A Polη-complemented XP30RO cells (Fig. 5d and Supplementary Fig. 7f). We also transfected WT, T457A, and K462R GFP-Polη into 293T cells and compared the level of their O-GlcNAc modification. Unlike T457A, K462R mutation did not impair

Polη O-GlcNAcylation (Fig. 5e), which is consistent with our hypothesis that K462 polyubiquitination happens after T457 O-GlcNAcylation. In line with this, knockdown of the CRL4[CDT2] components did not impair Polη O-GlcNAcylation either (Fig. 5f).

**T457A affects Polη TLS after cisplatin treatment.** Besides UV-induced CPD lesions, Polη is also known to participate in cisplatin-induced TLS, which is believed to contribute to the development of cisplatin resistance[12,19,21]. We found that cisplatin treatment enhanced the interaction between OGT and Polη (Fig. 6a), and promoted Polη O-GlcNAcylation (Fig. 6b), suggesting that Polη O-GlcNAcylation regulates cellular response to

cisplatin. As expected, the T457A-complemented cells displayed a much lower survival rate after cisplatin treatment when compared with WT Polη-complemented cells (Fig. 6c). We also examined Polη focus formation at different time points after cisplatin treatment. There are no significant differences in Polη focus formation at 24 h after cisplatin treatment between WT- and T457A-expressing cells (Fig. 6d). However, the reduction rate in proportion of GFP-Polη foci-positive cells at 34 and 46 h after drug removal was much slower in T457A-expressing cells than that of WT-expressing cells (Fig. 6d), indicating that T457A mutation might impair cellular TLS efficiency after cisplatin treatment through limiting Polη removal from damage sites. To test this possibility, we compared the RPA signals in WT and T457A Polη-complemented XP30RO cells after cisplatin

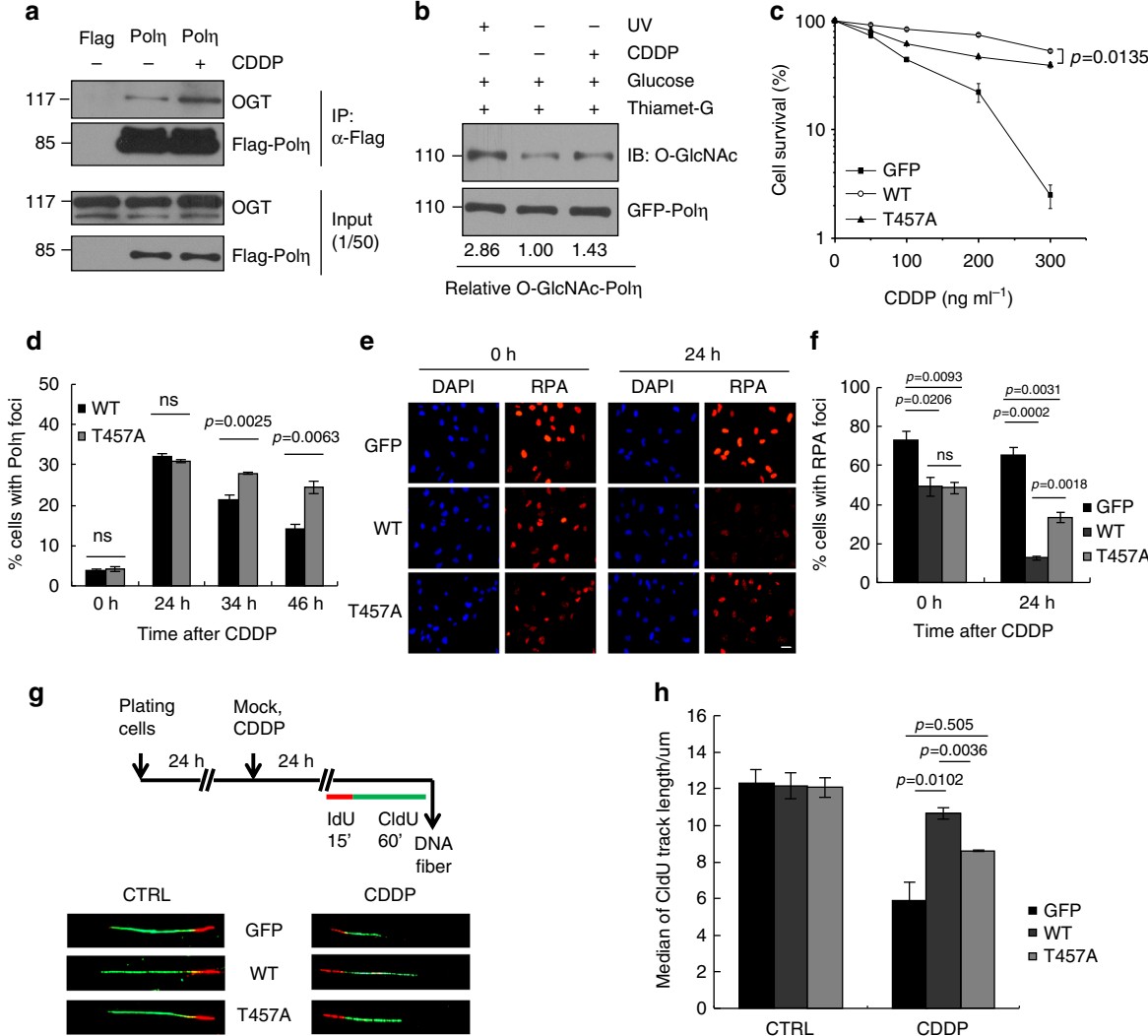

**Fig. 6** T457 affects Polη TLS efficiency after cisplatin treatment. **a** 293T cells transfected with Flag-Polη were treated with CDDP (10 μg ml⁻¹) for 2 h and changed to drug-free media for 2 h. The cell lysates were harvested for immunoprecipitation assay as in Fig. 1e. **b** 293T cells transfected with GFP-Polη were treated with Thiamet-G and glucose. Cells were treated with UVC (15 J m⁻²) irradiation or cisplatin (10 μg ml⁻¹) and harvested at 4 or 2 h later, respectively. The cell lysates were immunoprecipitated with GFP-Trap A and analyzed with anti-O-GlcNAc and anti-GFP antibodies. **c** XP30RO cells stably expressing GFP, WT, or T457A GFP-Polη were treated with cisplatin for 24 h and further incubated for 7–10 days. Cell survival was analyzed as in Fig. 3f. **d** U2OS cells transfected with WT or T457A GFP-Polη were treated with cisplatin (5 μg ml⁻¹) for 2 h followed by incubation with fresh media. At the indicated time points, the proportion of GFP-Polη expressing cells with >30 foci was determined. **e** Representative images of cells stained with DAPI and RPA2 after cisplatin treatment. XP30RO cells stably expressing GFP, WT, or T457A GFP-Polη were treated with cisplatin for 24 h and further cultured. At the indicated time points, cells were immunostained with anti-RPA2 antibody. Scale bars: 10 μm. **f** Quantification of the percentage of cells with >10 RPA2 foci. **g** Scheme of experimental design for replication fork progression in XP30RO cells stably expressing GFP, WT, or T457A GFP-Polη. After cisplatin (1 μg ml⁻¹) treatment for 24 h, representative DNA fibers were shown. **h** Median lengths of nascent replication tracts (labled with CldU) were given. Data represent means ± SEM from three independent experiments. ns not significant

treatment. The percentages of cells with RPA foci were similar between WT- and T457A-complemented cells immediately after cisplatin treatment (Fig. 6e, f). However, after 24 h recovery, the proportion of RPA foci-positive cells in WT Polη-expressing group (12.6%) was significantly less than that in the T457A mutant-expressing group (33.4%), supporting that T457A mutation impairs cisplatin-induced cellular TLS efficiency. We further performed DNA fiber analysis to investigate the direct effect of T457A mutation on bypass of cisplatin-induced DNA lesions in vivo. Cisplatin-treated cells were pulse-labeled with iodo-deoxyuridine (IdU) for 15 min followed by chlorodeoxyuridine CIdU for 60 min (Fig. 6g). Then the length of newly synthesized DNA strands (green fluorescent CIdU) was monitored. In the absence of cisplatin, the median fiber lengths in GFP, WT, or T457A Polη-expressing cell lines were similar (12.30, 12.17, and 12.07 μm, respectively; Fig. 6g, h), indicating that their replication fork rates were comparable. But after exposure to cisplatin, the median fiber length in T457A Polη-expressing cells (8.62 μm) was substantially shorter than that in WT Polη-expressing cells (10.65 μm) ($p = 0.0036$), although it was longer than that in GFP-complemented cells (5.90 μm; $p = 0.505$). These data provide direct evidence that inactivation of Polη O-GlcNAcylation at T457 impairs replication fork progression in vivo following cisplatin treatment.

## Discussion

Polη is the only known DNA polymerase that has an enlarged active site to accommodate the entire CPD or Pt-GG for the 3′ base to instruct correct nucleotide incorporation[19]. It can be recruited to stalled replication forks through interaction with mUb-PCNA[7,9,10,23]. Owing to its low fidelity on undamaged DNA, timely removal of Polη from replication forks when it is not required is presumably essential for genome stability[6]. Although Polη-reduced affinity for the DNA beyond the CPD might contribute to its dissociation after TLS[28], deubiquitination of mUb-PCNA by USP1[29] or USP10[30] can promote Polη displacement from replication forks. Notably, mUb-PCNA usually persists for many hours even after clearance of the damage and TLS polymerase foci[31]. Additionally, upon cisplatin exposure, efficient bypass of Pt-GG usually needs a second TLS polymerase to carry out primer extension after dCTP incorporation opposite the cross-linked bases by Polη[6,19,53]. This polymerase-switching between insertion and extension polymerases usually happens in the presence of mUb-PCNA. At present, how disassembly of Polη occurs in the persistence of mUb-PCNA still remains enigma.

Herein we demonstrated that human Polη undergoes O-GlcNAcylation. Through MS, we showed that Polη is mainly O-GlcNAcylated at residue T457. By monitoring the disappearance of CPDs in ssDNA[41] and the disappearance of RPA foci[44] after UV irradiation, we showed that T457A mutation does not affect the ability of Polη to bypass CPD lesions. Intriguingly, this mutation remarkably impairs disassembly of Polη from replication forks after TLS is finished, causing increased mutation frequency as well as UV sensitivity.

Considering that the p97–UFD1–NPL4 complex acts as a K48 ubiquitin-chain-dependent protein segregase to remove DNA-binding proteins at the sites of DNA damage[54], it is therefore tempting to speculate that O-GlcNAcylation at T457 is likely required for timely Polη ubiquitination and thereby p97-dependent dissociation from replication forks after TLS. Consistently, we found that Polη interacts with the CRL4^CDT2 complex, which is responsible for ubiquitination of multiple PIP degron-containing proteins[51]. Inactivation of Polη O-GlcNAcylation at T457 inhibits its association with this E3 complex leading to a distinct reduction in the level of Polη K48-

polyubiquitination. Additionally, based on quantitative MS analysis, several potential Polη ubiquitination sites regulated by T457 O-GlcNAcylation were identified. Among them, two T457-adjacent residues (K453 and K462) were mutated to assess their effects on Polη removal from replication forks. Only K462R manifested an epistasis relationship with T457A in restraining timely Polη dissociation from replication forks as well as causing an increased rate of mutagenesis and hypersensitivity after UV irradiation. Further analysis revealed that K462 can be poly-ubiquitinated through K48-linked ubiquitin chains on chromatin, echoing the previous report[52] that K462 is a potential CRL modification site. Interestingly, synergistic effects on reduction of Polη K48-linked polyubiquitination were not observed in the T457A/K462R double mutant compared with the K462R single mutant, suggesting that K462 polyubiquitination sequentially happens after T457 O-GlcNAc modification. In support of this, K462R mutation or depletion of the components of CRL4^CDT2 E3 ligase does not impair Polη O-GlcNAcylation at T457.

Besides CPDs, Polη is capable of replicating across Pt-GG adducts[6,18,19], which is believed to contribute to the development of cisplatin resistance[12,19,21]. We demonstrated an augmentation of Polη O-GlcNAcylation after cisplatin treatment, and inactivation of this modification due to T457A mutation sensitizes cells to cisplatin killing. Interestingly, cisplatin-induced RPA foci disappear more slowly in cells carrying T457A Polη compared to the WT counterparts, suggesting a TLS defect upon inactivation of Polη O-GlcNAcylation at T457. Mechanistically, through DNA fiber analysis, we confirmed that TLS is strongly reduced across cisplatin-damaged DNA after T457A mutation. This impact seems distinct from the effect of T457A mutation on TLS across CPDs. However, in light of the fact that Polη can finish bypass synthesis of CPD by its own, but fail to support complete bypass of Pt-GG[19], it is likely that the inability of T457A Polη to fulfill efficient TLS after cisplatin treatment might also be attributed to its restrained disassociation from replication forks.

In our study, we also showed that Polη O-GlcNAcylation is induced by both UV and cisplatin exposure. Given that OGT and O-GlcNAcylation can be enriched at the sites of DNA damage[38], it is conceivable that the upregulation of Polη O-GlcNAcylation occurs on chromatin, after it is recruited to stalled replication forks. Congruently, inactivation of Polη O-GlcNAcylation by T457A mutation does not impair Polη focus formation after UV and cisplatin treatment. Therefore, we speculate that a key function of Polη O-GlcNAcylation at T457 is to facilitate polymerase switching during TLS-mediated bypass of fork-blocking lesions and additionally help to dissociate low-fidelity TLS polymerases from replication forks after TLS, thereby restricting excessive mutagenesis (Fig. 7). This mechanism is apparently different from that of a recently identified phosphorylation of serine 687 on Polη, which is postulated to facilitate Polη departure from PCNA through bringing negative charges to the nuclear localization signal of Polη[55].

Recently, Polη sumoylation at K163 was identified, which was reported to be required to prevent under-replicated DNA[56]. Interestingly, through MS, we found that K163 residue can also be ubiquitinated, which is downregulated by T457A mutation. Therefore, it is yet to be determined how these different PTM modes interplay to collaboratively regulate Polη in vivo.

The upregulation of O-GlcNAcylation has been recognized as a common feature of cancer cells[57]. Recently, Polη is reported to play a vital role in improving ovarian cancer survival by enrichment of cancer stem cells after cisplatin treatment[21]. Based on our result that mutation of the major Polη O-GlcNAcylation residue T457 sensitizes cell to cisplatin exposure, it is reasonable to speculate that targeting Polη O-GlcNAcylation in ovarian cancer cells may have potential therapeutic benefits. Since Polη can carry

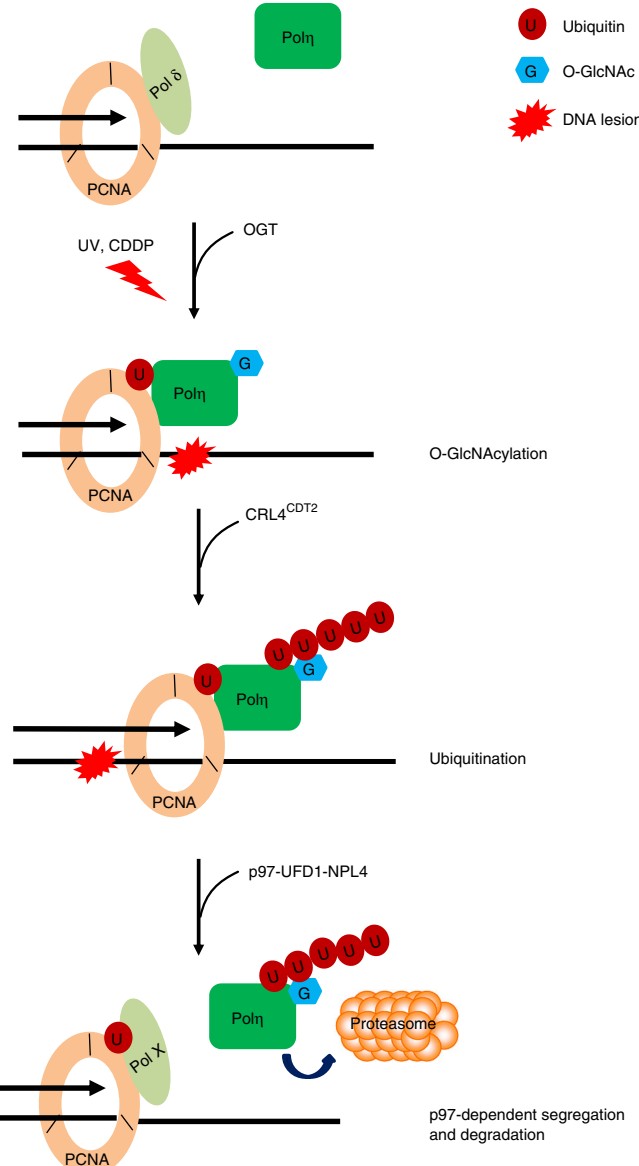

**Fig. 7** A proposed model depicting the role of Polη O-GlcNAcylation in its removal after TLS. Upon UV or CDDP exposure, Polη is recruited to stalled replication forks and gets O-GlcNAcylated by OGT. Once Polη's role in TLS is completed, O-GlcNAcylated Polη is ubiquitinated by CRL4^CDT2 E3 ligase complex. The polyubiquitinated Polη is then recognized by p97-UFD-NPL4 complex, resulting in its dissociation from replication forks and degradation

out other roles beyond TLS in vivo[58–61], it will be of great value to examine whether Polη O-GlcNAcylation also affects these processes. Additionally, considering that OGT and O-GlcNAcylation can be enriched at the sites of DNA damage[38], where other TLS polymerases and multiple other DDR factors can be recruited, it will be interesting to delineate whether O-GlcNAcylation also modulates the dynamic association/dissociation of those factors from the sites of damage in future studies.

In summary, our results reveal that loss of O-GlcNAcylation at T457 in Polη significantly decreases K48-linked polyubiquitination at adjacent K462, thereby restraining its timely removal from replication forks and subsequent polymerase switching during and after TLS. Given the key role of Polη in TLS and cellular resistance to chemotherapy, O-GlcNAcylation adds a further layer of regulation that controls TLS and genome stability in vivo.

## Methods

**Plasmids and reagents**. Human Polη cDNA was cloned in pEGFP-C3 (Clontech) or pCMV-2xFlag-SBP (modified from pCMV-3xFlag-myc) to generate eGFP or Flag fusion proteins. Different O-GlcNAc mutants including T457A, 3A (T457A, S458A, S459A), 5A (T457A, S458A, S459A, S466A and T471A), K453R, K462R, and T457A/K462R were constructed based on pEGFP-C3-Polη or pCMV-2xFlag-SBP-Polη by site-mutated PCR amplification. Flag-NPL4 and Flag-UFD1 were gifts from Dr. Wei Li (Institute of Zoology, Chinese Academy of Science). Flag-CDT2 and Myc-Cul4A were gifts from Dr. Hengyu Fan (Zhejiang University). pcDNA3-Flag-DDB1 was purchased from Addgene. pET-28a-Polη was a gift from Dr. Wei Yang (NIDDK, National Institutes of Health). For generating internal deletion mutants, cDNA fragments of Polη or OGT genes were PCR amplified and cloned into pCMV-3xFlag or pCMV-Myc, respectively. Anti-Flag M2 agarose affinity gel (A2220) was purchased from Sigma (St Louis, MO). GFP-Trap A beads (gta-20) were from Chromotek. Antibodies sources were as follows: mouse anti-Flag (F1804, 1:1000) from Sigma (St Louis, MO); anti-CPD (CAC-NM-DND-001, 1:4000) from Cosmo Bio Co (Tokyo, Japan); anti-Polη (ab17725, 1:1000) from Abcam and Dr. Simone Sabbioneda (Istituto di Genetica Molecolare, Italy); anti-RPA32 (ab2175, 1:800), p97 (ab11433, 1:2000), and O-linked β-N-acetylglucosamine (O-GlcNAc, ab2739, 1:2000) from Abcam; anti-HA (902302, 1:2000) from BioLegend; anti-Myc (MMS-150R-500, 1:1000) from Covance; anti-His (HT501-02, 1:1000) from Beijing TransGen Biotech Co., Ltd; anti-H3.1 (P30266, 1:2000) from Abmart; anti-β-Tubulin (AbM59005-37-PU, 1:4000) from Beijing Protein Innovation (Beijing, China); anti-CDT2 (A300-948A, 1:500) and Cullin 4A (A300-739A, 1:1000) from Bethyl; anti-DDB1 (NB100-625, 1:1000) from Novus Biologicals; anti-GFP (sc-8334, 1:500), OGT (sc-32921, 1:1000), and ubiquitin (sc-8017, 1:1000) from Santa Cruz Biotechnology. Ubiquitin chain-specific Lys48 antibody (4298, 1:200) was from CST (Cell Signaling Technology). Alexa Fluor-conjugated secondary antibodies were from Invitrogen. Uncropped immunoblots of the main figures are shown in Supplementary Figs. 8–10.

**Cell culture and reagents**. Human U2OS and 293T cells were obtained from the American Type Culture Collection (Rockville, MD). SV40-transformed MRC5 and XP30RO cells were kindly provided by Dr. Alan Lehmann[5]. The XP30RO-Polη cell line (a Polη-deficient XP30RO cell line to express SFB-tagged Polη under the control of a tetracycline-inducible promoter) was kindly provided by Dr. Jun Huang[42]. These cell lines were grown in Dulbecco's modified Eagle's medium medium supplemented with 10% fetal bovine serum. All cells were grown at 37 °C in the presence of 5% CO$_2$ if not specified. All cells were tested for mycoplasma contamination using the Lonza Mycoplasma Kit. For transient transfection experiments, cells were transfected with the indicated constructs, using Vigofect (Vigorous Biotechnology) following the manufacturer's protocols. For RNAi experiments, cells were transfected with small interfering RNAs (siRNAs) purchased from GenePharma (Shanghai, China) using RNAiMAX (Invitrogen) according to the manufacturer's instruction and analyzed 72 h later. The gene-specific target sequences were as follows: DDB1-1 (GCAAGGACCUGCU-GUUUAU), DDB1-2 (GGCCAAGAACAUCAGUGUG), CDT2 (GAAUUAUA-CUGCUUAUCGA), p97-1 (CCCAAGAUGGAUGAAUUGCAGUUGU), p97-2 (AACAGCCAUUCUCAAACAGAA), Cul4A (GAAGCUGGUCAUCAAGAAC), OGT-1 (UAAUCAUUUCAAUAACUGCUUCUGC), and OGT-2 (GAUUAAGCCUGUUGAAGUC). The negative control (siNC) sequence (UUCUCCGAACGUGUCACGU) was also obtained from GenePharma.

**Establishment of Polη rescue cell lines**. Lentiviral vectors expressing a series of GFP-tagged Polη were constructed and transfected into 293T cells with packaging plasmids pCMV-DR and pMD2-VSVG to produce viral particles[62]. The lentiviral particles were then used to infect XP30RO cells with the addition of 8 μg ml⁻¹ polybrene (Sigma-Aldrich). The stable cells were selected by flow cytometry, and individual clones were isolated. The expression levels of Polη were confirmed by western blotting with antibodies against GFP or Polη, as indicated.

**Higher-energy collision dissociation MS**. HEK293T cells expressing 2xFlag-Polη were incubated with 5 μM Thiamet-G for 24 h and 30 mM glucose for 3 h. Cells were harvested and lysed with HEPES buffer (50 mM HEPES pH 7.5, 150 mM NaCl, 1 mM EDTA, 1 mM EGTA, 10% glycerol, 1% Triton X-100, 25 mM NaF, 10 μM ZnCl$_2$). The whole-cell lysates were subjected to IP using anti-Flag M2 agarose. The bound proteins were eluted with 0.1 M Glycine (pH 2.5) and neutralized by adding Tris (pH 8.5). The elutions were precipitated using ice-cold acetone and resuspended in 8 M urea and 100 mM Tris (pH 8.5). After reduction with TCEP (Tris(2-Carboxyethyl)-Phosphine) (5 mM) for 20 min and alkylation with iodoacetamide (10 mM) for 15 min at room temperature, the samples were diluted to 2 M urea with 100 mM Tris (pH 8.5), and each divided into four aliquots for Trypsin, Glu-C, or Elastase digestion (1:50 enzyme:substrate, 25 °C for Glu-C and 37 °C for Trypsin and Elastase) overnight. The digestions were quenched with 5% formic acid and pooled together. The LC-MS/MS analysis was performed on an Easy-nLC 1000 UPLC (Thermo Fisher Scientific) coupled to a Q Exactive mass spectrometer (Thermo Fisher Scientific). Peptides were loaded on a precolumn (75 μm internal diameter, 8 cm long, packed with ODS-AQ 12 nm-10 μm beads from YMC Co., Ltd), and separated on an analytical column (75 μm internal diameter, 11 cm long, packed with

Luna C18 3 μm-100 Å resin from Phenomenex) using an acetonitrile gradient from 0 to 30% in 55 min and 30–80% in another 10 min at a flow rate of 300 nl min$^{-1}$. Spectra were acquired in a data-dependent mode: the 15 most intense precursor ions from each full scan (Resolution 70,000) were isolated for HCD MS2 (Resolution 17,500) at NCE 27 with a dynamic exclusion time of 20 s. Precursors with 1+ or unassigned charge states were excluded. For peptide identification, the MS2 spectra were searched against an EPI-IPI human database (forward and reversed sequences) using Prolucid with 50 ppm mass accuracy for both precursor and fragment ions, with carbamidomethylation on cysteine as fixed modification and GlcNAc (203.079373) on serine, threonine, or tyrosine as differential modification[63]. Search results were filtered using DTASelect 2.0 with a 5% false discovery rate cutoff at the spectral level[64]. The GlcNAc-modified peptide spectrum was labeled using pLabel, requiring 20 ppm mass accuracy for fragment ions[65].

**Chromatin fraction isolation**. Cells were lysed in CSK-100 buffer (100 mM NaCl, 300 mM sucrose, 3 mM MgCl$_2$, 10 mM PIPES pH 6.8, 1 mM EGTA, 0.2% Triton X-100) containing protease inhibitors at 4 °C for 15 min. Chromatin-associated proteins were released from the pellets by treatment with lysis buffer (50 mM HEPES pH 7.5, 50 mM NaCl, 0.05% SDS, 2 mM MgCl$_2$, 10% Glycerol, 0.1% Triton X-100, 10 units of Benzonase Nuclease) containing protease inhibitors at 4 °C overnight. The supernatants were separated by SDS-PAGE and detected by immunoblotting with the indicated antibodies[9].

**Immunofluorescence**. Cells were cultured on glass coverslips. Briefly, cells were treated with 0.5% Triton X-100 for 5–30 min before fixing in 4% paraformaldehyde. Then the cells were incubated with 5% fetal bovine serum and 1% goat serum for 1 h followed by incubation with anti-RPA2 for 45 min. After staining with secondary antibodies (Alexa Fluor 568; Molecular Probes) for 45 min, cover slips were mounted in Vectashield mounting medium (Vector Laboratories) containing the nuclear stain 4, 6-diamidino-2-phenylindole (DAPI). Images were acquired with a Leica DM5000 (Leica) equipped with HCX PL S-APO 63 × 1.3 oil CS immersion objective (Leica) and processed with Adobe Photoshop 7.0[41].

For quantitative analysis of UV-induced Polη focus formation, U2OS cells transfected with GFP-Polη were treated with UVC (15 J m$^{-2}$) and fixed with 4% paraformaldehyde 8–24 h later after UV irradiation[41]. Images were acquired using a Leica DM5000 (Leica) equipped with HCX PL S-APO 63 × 1.3 oil CS immersion objective (Leica). A minimum of 200 nuclei was analyzed for each treatment.

**Detection of unreplicated CPDs in UV-treated cells**. Detection of CPDs in single-stranded DNA templates was performed as previously described[41] with some modifications. Briefly, cells cultured on coverslips were irradiated with 0 or 5 J m$^{-2}$ UVC light. Four hours later, cells were treated with 1% Triton-X100 in phosphate-buffered saline (PBS) for 2 min and subsequently fixed in 2% Formaldehyde/PBS containing 0.5% Triton-X100 for 15 min at room temperature. Unreplicated photoproducts were identified in non-denatured DNA using primary mouse monoclonal antibody against CPDs (TDM2, CosmoBio). After incubation with secondary Alexa Fluor 568-labeled goat-anti-mouse antibodies (Molecular Probes, Inc.), nuclei were stained with DAPI. Images were analyzed by fluorescent microscopy.

**Cell cycle analysis**. U2OS cells transfected with WT or T457A GFP-Polη were treated with UVC (15 J m$^{-2}$) and harvested by trypsinization 24 h later after UV irradiation. Cells were washed twice with PBS and fixed in ice-cold 70% ethanol overnight. After treatment with RNase (suspended in PBS, containing 0.1% Triton X-100 and 1% BSA) at room temperature for 20 min, cells were incubated with propidium iodide for 30 min in the dark followed by termination with PBS. Cell cycle distribution was analyzed by a FACSCalibur flow cytometer (BD Biosciences).

**Co-IP and western blotting**. HEK293T cells transfected with Flag-Polη or GFP-Polη were harvested and lysed with HEPES buffer (50 mM HEPES pH 7.5, 150 mM NaCl, 1mM EDTA, 1 mM EGTA, 10% glycerol, 1% Triton X-100, 25 mM NaF, 10 μM ZnCl$_2$). The whole-cell lysates were immunoprecipitated with either anti-Flag M2 agarose or GFP-Trap A beads (Chromotek)[66]. Samples were separated by SDS-PAGE and detected by immunoblotting with the indicated antibodies. To confirm Polη O-GlcNAcylation, 293T cells expressing Flag-Polη were lysed with 1xSDS lysis buffer (50 mM Tris-HCl pH 6.8, 100 mM DTT, 2% SDS, 10% glycerol) at 95 °C for 15 min. The denaturated lysates were centrifuged and the supernatant was diluted with HEPES buffer (1:14) followed by IP with anti-Flag M2 beads. The immunoprecipitated products were separated by SDS-PAGE and analyzed with anti-O-GlcNAc antibody. To confirm that protein association happens on chromatin, HEK293T cells transfected with the indicated plasmids were harvested and permeabilized by CSK-100 buffer (100 mM NaCl, 300 mM Sucrose, 3 mM MgCl$_2$, 10 mM PIPES pH 6.8, 0.2% Triton X-100) at 4 °C for 15 min. The lysates were centrifuged and the supernatant was collected as the soluble fraction. The pellet was then lysed with buffer (50 mM HEPES, 50 mM NaCl, 10% glycerol, 10 uM ZnCl$_2$, 2 mM MgCl$_2$, 0.05% SDS, 0.1% Triton X-100) at 4 °C overnight before centrifugation. The supernatant was diluted with buffer (50 mM HEPES, 50 mM NaCl, 10% glycerol, 10 uM ZnCl$_2$, 0.1% Triton X-100) (1:10) followed by IP with anti-Flag M2 beads. For denatured chromatin fraction IP, the pellet was lysed with 1xSDS lysis buffer as above. The immunoprecipitated products were separated by SDS-PAGE and analyzed with the indicated antibodies. The relative O-GlcNAcylation or polyubiquitination levels of Polη in each sample were represented, with the O-GlcNAcylation or polyubiquitination level of the control sample set to 1 (100%). The gray densities of the O-GlcNAcylation or polyubiquitination signals and those of unmodified Polη were determined by the Photoshop software (Adobe Systems Incorporated, USA).

**In vivo ubiquitination assay**. HEK293T cells were transfected with the indicated OGT, DDB1, CDT2, or p97 siRNAs twice, and 48 h after the first transfection, cells were further transfected with HA-Ub and a series of Flag- or GFP-tagged Polη plasmids. The cell lysates were immunoprecipitated with Flag M2 agarose or GFP-Trap A beads followed by western blotting with anti-Ubiquitin and anti-HA antibodies to detect the ubiquitination of Polη.

**In vitro O-GlcNAcylation assay**. pET-28a-WT or T457A Polη (with kanamycin resistance) was co-transformed with pGEX-4T-2-OGT (with ampicillin resistance) into E. coli Transetta (DE3) cells. Single clones selected on ampicillin/kanamycin plate were grown at 37 °C in Luria–Bertani (LB) medium until they reached OD$_{600}$ = 0.6, then isopropy β-D-1-thiogalactopyranoside (IPTG) (0.4 mM) was added and cultured at 16 °C overnight. His-tagged Polη was affinity-purified using Ni-NTA Agarose (Qiagen) and resolved by SDS-PAGE followed by immunoblotting with antibodies against O-GlcNAc and His[67].

**In vitro ubiquitination assay**. Briefly, Flag-CDT2, Myc-Cul4A, Myc-DDB1, and Myc-RBX1 plasmids were co-transfected into HEK293T cells. CRL4$^{CDT2}$ complexes immobilized on Flag M2 agarose beads were washed with HEPES buffer followed by elution with Flag peptide. For the in vitro ubiquitination assay, purified WT or T457A His-Polη proteins were incubated with the eluted CRL4$^{CDT2}$ complex in a ubiquitin ligation reaction mixture (final volume, 10 μl) containing 50 mM Tris-HCl (pH 7.5), 5 mM MgCl$_2$, 10 nM okadaic acid, 2 mM ATP, 0.5 mM DTT, 5 ug ubiquitin (Boston Biochem), 60 ng E1-UbE1 (Boston Biochem), and 300 ng E2-UbcH5c (Boston Biochem) at 30 °C for 2 h. The reaction was terminated by boiling for 5 min in a SDS sample buffer, and the proteins were resolved by SDS-PAGE, followed by immunoblotting with the indicated antibodies[68].

**Quantitative MS of Polη ubiquitination**. HEK293T cells expressing WT or T457A 2xFlag-Polη and HA-Ub were harvested 48 h after plasmids' transfection and lysed with HEPES buffer (50 mM HEPES pH 7.5, 150 mM NaCl, 1 mM EDTA, 1 mM EGTA, 10% glycerol, 1% Triton X-100, 25 mM NaF, 10 μM ZnCl$_2$). WT or T457A Polη immunoprecipitated using anti-Flag M2 agarose were eluted by 3xFlag peptide and reduced with 5mM DTT at 80 °C for 10 min and alkylated with 10 mM iodoacetamide in dark at room temperature for 30 min. Proteins were resolved by SDS-PAGE (10%) and stained with Coomassie Blue G250. Then the gel lanes were cut into two parts and digested in-gel with 12.5 ng μl$^{-1}$ of trypsin at 37 °C for 14 h. After extraction from the gel bands with 50% acetonitrile (AcN) and 5% formic acid, the peptide mixtures were dried and dissolved with sample loading buffer (1% formic acid, 1% acetonitrile) for MS analysis.The peptide mixtures were separated and analyzed by ultra performance LC (nano Acquity Ultra Performance LC, Waters, Milford, MA) and tandem MS/MS (LTQ Orbitrap Velos, Thermo Fisher Scientific, Waltham, MA). The samples were loaded onto the column and eluted with a 1 h gradient with 1–35% of buffer B (buffer A: 2% AcN and 0.1% formic acid; buffer B: 100% AcN 0.1% formic acid; flow rate, ~300 nl min$^{-1}$). Survey scans were performed in the Orbitrap analyzer at a resolution of 30,000 at m/z 400. For low molecular weight part, the 20 most intense peptide ions with charge state ≥2 were subjected to fragmentation via collision-induced dissociation in the LTQ (2000 AGC target, 50 ms maximum ion time). In order to increase sensitivity of K462 GG-peptide, selective reaction monitoring strategy was applied as previously described[69]. Targeted ions (m/z = 711.01, m/z = 701.01) and the three most intense peptide ions were detected in MS/MS analysis.

The raw data files were searched by the Maxquant (version 1.5.6.5) against the Swiss-Prot reviewed database (version released in 2016.12 for human). Searching parameters consisted of full tryptic restriction, fixed modification of Cys (+57.0215 Da), and variable modifications of oxidized Met (+15.9949 Da) and di-Gly-lysine (114.0429Da). The mass tolerance was set to 20 ppm for precursor ions and 0.5 Da for product ions. The peptide intensity for quantification was manually analyzed by extracted ion chromatogram.

**DNA fiber assay**. Cells were treated with CDDP (2 μg ml$^{-1}$) for 24 h. Before harvesting, cells were pulse-labeled with the modified thymidine analogues IdU (100 μM) for 15 min, followed by CIdU (100 μM) for 60 min. Cells were collected, lysed, and spread on microscope slides to obtain single-molecule DNA fibers. The labeled replication tracts were detected with primary antibodies against IdU (BD Biosciences, anti-BrdU clone B44) and CIdU (Abcam, anti-BrdU BU1/75/(ICR1)) and secondary antibodies (Invitrogen, Alexa Fluor 488 goat anti-rat and Alexa Fluor 568 goat anti-mouse). Fibers were imaged using a Leica DM5000B microscope. Length of 10–150 DNA fibers were measured using the ImageJ software from 2–3 independent experiments. p-Values were obtained from the Mann–Whitney test[66].

**supF shuttle vector-based mutagenesis assay**. Mutation frequencies were measured using the supF shuttle vector system[49], which measures TLS activity in mammalian cells[46]. 293T cells transfected with the indicated siRNA oligos or XP30RO-lenti-GFP-Polη stable cells were transfected with a UVC-irradiated (400 J m$^{-2}$) pSP189 reporter plasmid. Forty eight hours later, the pSP189 plasmid was retrieved from the cells using a DNA Miniprep Kit (Tiangen, China). The purified plasmid was digested with DpnI and transformed into the MBM7070 bacterial strain. The transformed MBM7070 cells were grown on LB plates containing 200 μM IPTG, 100 μg ml$^{-1}$ X-gal, and 100 μg ml$^{-1}$ ampicillin. The mutation frequency in the supF-coding region was determined by enumerating the ratios of blue (WT) and white (mutant) colonies. The pSP189 plasmid and MBM7070 strain were gifts from Dr. M. Seidman.

**Cell survival assay**. Cells were seeded into 6-cm dishes (~200 cells/dish) in triplicate and allowed to adhere for 5 h. The cells were then treated with the indicated doses of CDDP for 24 h at 37 °C or UVC irradiation. After treatment, cells were further incubated in complete medium for 7–10 days. For UVC irradiation, the cells were cultured along with caffeine (0.4 mM) for 30 min prior to treatment and then incubated in complete medium supplemented with caffeine (0.4 mM) for 7–10 days. Colonies were fixed and counted. The survival of genotoxin-exposed cells was determined by relating the cloning efficiency to that of an untreated control[70].

**Statistical analysis**. All statistical tests were determined with a two-sided Student's t-test using the PRISM software (Graphpad Software Inc.) unless otherwise noted. p-Values were rounded to four decimal places and differences were considered as statistically significant when $p < 0.05$.

**Data availability**. All data supporting the findings of this study are available from the corresponding authors on request.

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

## Acknowledgements

The authors thank Drs. Alan Lehmann, Simone Sabbioneda, Jun Huang, Wei Li, Hengyu Fan, and Jun Dou for reagents; Dr. Wei Yang, Dr. Yungui Yang, and Dr. Wei Li for helpful discussion; and Miss Qing Meng for cell sorting through FACS. This work was supported by NSFC81630078, 91754204, MOST2017YFC1001001, 2013CB911201 and 2014CB84980001, NSFC31471331, 31670822, 31570816, 91519324, 81371415, 31470784, and 31701227; CAS Strategic Priority Research Program XDB14030300; the State Key Laboratory of Membrane Biology; CNU interdisciplinary project; and Beijing Nova Program Interdisciplinary Cooperation Project (Z161100004916042).

## Author contributions

X.M. performed most of the experiments with help from H.L. and J.L.; Y.W. performed ubiquitination mass spectrometric analysis under the supervision of P.X.; Y.-H.D. performed O-GlcNAcylation mass spectrometric analysis under the supervision of M.-Q.D.; H.S. performed DNA fiber assay; Y.Y. performed cell survival assay; C.S., M.H., Y.L., and Y.T. constructed Polη mutant plasmids; Y.Z. provided discussion and proofread the manuscript. X.M., T.-S.T., and C.G. designed the whole project, analyzed the data, and wrote the manuscript with input from all authors.

## Additional information

**Competing interests:** The authors declare no competing financial interests.

