## [Peer Review File · Nature Communications]

Reviewers' comments:

Reviewer #1 (Remarks to the Author):

In their paper, Ma and colleagues identify a new modification for the Translesion synthesis polymerase pol η . They find that pol η is O-GlcNAcylated on T457 and this modification is important for its displacement after UV irradiation and cisplatin treatment. In particular, they find that inactivation of T457 results in decreased K48 linked polyubiquitylation of the polymerase. This in turn led to a reduced removal of the polymerase by p97 from the fork after damage. This increased accumulation of the polymerase also causes increased mutagenesis, hinting to persistent or unregulated error-prone TLS.

The authors further describe how T457 is required for efficient ubiquitylation of K462 by the CRL4 E3 complex, characterizing the later stages of the TLS polymerase switch.

The paper is very convincing and describes a completely new regulatory modification that controls pol η dependent translesion synthesis. Furthermore, the authors describe the involvement of the CUL4A complex in pol η ubiquitylation, a role that so far was described only in *C.elegans*. Overall, I find the story appealing and the results clear. For this reason, I think the paper is of interest for the readers of Nature Communications and it suitable for publication.

The only major issue I have is that the authors infer an increased retention of pol η T457A mutant only by quantifying the foci that the polymerase forms in the nuclei. Although focal recruitment can be used to cursory assess chromatin accumulation it not the correct readout to evaluate chromatin binding. The authors should provide a direct evidence of the increased retention of the polymerase by performing a chromatin fractionation experiment over time. Such an experiment will also provide the material to analyse if the binding to the chromatin of other TLS polymerases is affected and if the T457A mutation is indeed blocking the polymerase switch or termination of TLS bypass, as the authors suggest.

Minor Issues:

Fig2A: Quantify the amount of O-GlcNAc as the difference are not very big, especially in the case of the PIP mutant (bottom blot).

Fig2E: Pol η does not remove CPDs, it only bypasses them and I am surprised that the authors used it to test if T457A was active. At best, this indicates that the repair of CPD is working in the cells, maybe by combining the efficiency of NER repair and TLS bypass. Ideally, the authors should provide a direct evidence that T457A is active considering that the mutation is located close to the PAD domain.

Fig 3F/6C: Survival plot should be presented in log scale

Fig4B: Quality of the blot is very low with hardly any pol η visible.

Fig4F: Less T457A pol η was immunoprecipitated in the siP97 sample, so the reduction in Ubiquitin chains could be related to the different amounts of pol η

Fig6B: The differences mentioned by the authors are not as evident as the ones shown before. A quantitation of the blot would be beneficial here and in general where the authors state that there is a difference in O-GlcNAc band intensity.

Reviewer #2 (Remarks to the Author):

This is an interesting paper showing that pol η is modified by O-GlcNAcylation on Thr457, this modification increases after DNA damage and stimulates the Cdt2-mediated poly-ubiquitination of pol η on Lys462. Biological consequences of mutating these sites are presented and the paper provides a novel and compelling story. A couple of extra experiments are needed and a few minor amendments.

I.56 and I.330 The paper of Biertumpfel et al (2010) on the structure of pol eta suggests that dissociation after TLS may be an intrinsic property of pol eta because of reduced affinity for the DNA beyond the CPD. This should be mentioned here.

I.94 Although the data in Fig 1b are convincing, it should be pointed out that only a very small fraction of pol eta (<1%) is bound to OGT

I.143-146 The PIP box at 707-708 is well established, whereas that at 443-444 is controversial. In addition F443-445 is quite close to T457, so the effect of mutating these residues may be more direct. Have the authors mutated the two PIP boxes separately? If not, these data should probably be omitted.

I.147. The increase after UV does not look that great. Some quantitation and indication of reproducibility should be indicated. Also in view of the hypotheses proposed in the paper, it is important to see a time-course of the GlcNAcylation after UV.

I.174-175 FACS analysis should be provided to show whether the increased foci formation is the result of a higher proportion of cells in S phase.

I.198-199. In what cells were these mutagenesis experiments carried out? XP30RO?

Fig 3f should be plotted as log (survival) against dose. Was caffeine included in the post-irradiation incubation?

Fig 4 – a few questions. In a, what is the upper band in the siDDB1 lane. In b, why are the FLAG-pol eta bands so weak, compared to other panels, eg c, d, e? In panels c,d please explain exactly how the ratio was calculated. Why are panels e and f on the right and g and h on the left. This is confusing.

I.277-278. UV survival data for K462R should also be provided.

I.294, Fig 6c. Did the transfectants express wt and mutant pol eta at the same level?

I.314 This should be reworded to indicate substantial rescue in T457A cells but not up to wt levels.

I. 332 Please delete “for the first time”. Unnecessary and irritating.

I.792, fig 1f. Indicate meaning of SE and LE.

Reviewer #3 (Remarks to the Author):

DNA Polη plays a key role in Translesion DNA Synthesis (TLS) . With an affinity purification approach, Ma et al. identified OGT as a functional partner of DNA Polη. The authors further characterized that Polη is O-GlcNAcylated at T457 via the interaction with OGT. This novel posttranslational modification suppressed the CRL4CDT2-dependent ubiquitination at K462, thus impaired TLS. Overall, the manuscript may reveal a novel molecular mechanism underlying TLS. However, compelling evidence is needed to justify the major conclusions in this manuscript, and current form is too preliminary to warrant publication.

Major points:

1: Little evidence indicates that O-GlcNAcylation is involved in TLS. The author cited studies from other groups to show the biological function of O-GlcNAcylation in DNA damage repair. However, none of these studies directly address the significance of O-GlcNAcylation in TLS. The authors need

to establish the function of OGT or OGA in TLS before characterizing the possible O-GlcNAcylation sites in Polη.

2, In the manuscript, the authors used overexpression approach to examine the O-GlcNAcylation of Polη. However, the authors need to provide the evidence of endogenous O-GlcNAcylation on T457 of Polη.

3. If OGT directly interacts with Polη, it is essential for the authors to map the interaction regions, and perform the structure-function analysis.

4. The authors use the T457A mutation to claim the loss of O-GlcNAcylation, which is not accurate. As O-GlcNAcylation can compete with phosphorylation in many biological processes, this Thr residue may also be phosphorylated. The phenotype of the T457A mutation may result from the loss of phosphorylation.

5. The T467A mutation may simply abolish the consensus motif for CRL4CDT2-dependent ubiquitination. The phenotype may not even result from the loss of O-GlcNAcylation or phosphorylation per se. I strongly suggest the authors to use in vitro biochemistry assays to justify the major conclusions. A simple point mutation in the cell context may generate a lot of artifacts.

We appreciate that the paper was deemed to be of conceptual interest by reviewers, and we are also grateful to the reviewers' comments and constructive suggestions that certainly help us to improve greatly the quality of our manuscript. To address the reviewers' concerns, we have performed a number of experiments accordingly and made extensive revision in our revised manuscript.

Our point-by-point responses are listed below:

Reviewer #1 (Remarks to the Author)

In their paper, Ma and colleagues identify a new modification for the Translesion synthesis polymerase pol η . They find that pol η is O-GlnN-acylated on T457 and this modification is important for its displacement after UV irradiation and cisplatin treatment. In particular, they find that inactivation of T457 results in decreased K48-linked polyubiquitylation of the polymerase. This in turn led to a reduced removal of the polymerase by p97 from the fork after damage. This increased accumulation of the polymerase also causes increased mutagenesis, hinting to persistent or unregulated error-prone TLS.

The authors further describe how T457 is required for efficient ubiquitylation of K462 by the CRL4 E3 complex, characterizing the later stages of the TLS polymerase switch.

The paper is very convincing and describes a completely new regulatory modification that controls pol η -dependent translesion synthesis. Furthermore, the authors describe the involvement of the CUL4A complex in pol η ubiquitylation, a role that so far was described only in *C.elegans*. Overall, I find the story appealing and the results clear. For this reason, I think the paper is of interest for the readers of Nature Communications and it is suitable for publication.

The only major issue I have is that the authors infer an increased retention of pol η T457A mutant only by quantifying the foci that the polymerase forms in the nuclei. Although focal recruitment can be used to cursorily assess chromatin accumulation, it is not the correct readout to evaluate chromatin binding. The authors should provide direct evidence of the increased retention of the polymerase by performing a chromatin fractionation experiment over time. Such an experiment will also provide the material to analyse if the binding to the chromatin of other TLS polymerases is affected and if the T457A mutation is indeed blocking the polymerase switch or termination of TLS bypass, as the authors suggest.

Response: We really appreciate the reviewer's thoughtful comments. We have performed a chromatin fractionation experiment based on the reviewer's suggestion. In line with the result from Pol η foci analysis, mutation of T457 to A substantially decreased the displacement of Pol η from chromatin at 24 h after UV irradiation, with the amount of chromatin-bound T457 Pol η showing a ~2-fold increase over that in WT Pol η at 24 h post-UV. However, it did not obviously affect the level of chromatin-bound Pol η at 8 h after UV irradiation (Supplementary Fig. 2a). Therefore, T457A mutation caused an increased retention of Pol η at 24 h post-UV. However, due to the lack of suitable antibodies for other TLS polymerases, it is not feasible to determine

whether their binding to the chromatin are also affected.

Minor Issues:

Fig2A: Quantify the amount of O-GlcNAc as the difference are not very big, especially in the case of the PIP mutant (bottom blot).

Response: We have performed the quantification following the reviewer's suggestion. The result showed that glucose and Thiamet-G treatment caused an obvious increase in Polη O-GlcNAcylation, which was about 3.19 fold of that in control. UV treatment further increased Polη O-GlcNAcylation to about 8.21 fold of that in control (Fig. 2a, top panel). As for the original bottom blot, the reviewer #2 pointed out that although the PIP box at 707-708 is well established, whereas that at 443-444 is controversial, and suggested to delete this data. We have followed the suggestion to delete the bottom blot panel in the revised version.

Fig2E: Polη does not remove CPDs, it only bypasses them and I am surprised that the authors used it to test if T457A was active. At best, this indicates that the repair of CPD is working in the cells, maybe by combining the efficiency of NER repair and TLS bypass. Ideally, the authors should provide a direct evidence that T457A is active considering that the mutation is located close to the PAD domain.

Response: We apologize for not clearly presenting this part in our original submission. In fact, this experiment mainly measured the unreplicated CPDs in single-stranded DNA templates, namely, unreplicated CPDs in non-denatured DNA. The cells were fixed and stained under non-denatured conditions for CPD after UVC treatment. This result does not measure CPD removal (or CPD repair), but reflects TLS bypass across CPD lesions as reported (Temviriyankul P, et al. DNA Repair. 11: 550-558. 2012). We found that although unreplicated CPDs persisted in the GFP-expressing XP30RO cells, they were undetectable in the XP30RO cells expressing either WT or T457A Polη at 4 h after UV irradiation, suggesting that efficient bypass across CPDs happens in either WT or T457A Polη-expressing XP30RO cells. Additionally, given that the disappearance of RPA foci after UV is consistent with the operation of TLS (Diamant N., et al. Nucleic Acids Research. 40:170-80. 2012), the comparable and scanty RPA signals in WT or T457A Polη-expressing cells at 24 h post UV irradiation (Fig. 2f, g) also support that T457A mutation does not compromise Polη's ability to bypass CPD lesions.

Fig 3F/6C: Survival plot should be presented in log scale

Response: The survival plots in Fig 3F/6C have been presented in log scale.

Fig4B: Quality of the blot is very low with hardly any polη visible.

Response: Thanks for pointing out this. We have included a longer exposure of polη blot in the revised Fig. 4B.

Fig4F: Less T457A polη was immunoprecipitated in the siP97 sample, so the reduction in Ubiquitin

chains could be related to the different amounts of polη

Response: We have repeated the experiment and shown that when similar amounts of T457A and WT Polη were immunoprecipitated in the siP97 samples, WT Polη still exhibited a significantly higher level of polyubiquitination compared to T457A. This result excludes the possibility that the reduction in Ubiquitin chains could be related to the different amounts of Polη.

Fig6B: The differences mentioned by the authors are not as evident as the ones shown before. A quantitation of the blot would be beneficial here and in general where the authors state that there is a difference in O-GlcNAc band intensity.

Response: As per the suggestion, we have quantified the blots, and the data showed that cisplatin treatment led to a 43% increase in the level of Polη O-GlcNAcylation (Fig. 6b).

Reviewer #2 (Remarks to the Author):

This is an interesting paper showing that pol eta is modified by O-GlcNAcylation on Thr457, this modification increases after DNA damage and stimulates the Cdt2-mediated poly-ubiquitination of pol eta on Lys462. Biological consequences of mutating these sites are presented and the paper provides a novel and compelling story. A couple of extra experiments are needed and a few minor amendments.

l.56 and l.330 The paper of Biertumpfel et al (2010) on the structure of pol eta suggests that dissociation after TLS may be an intrinsic property of pol eta because of reduced affinity for the DNA beyond the CPD. This should be mentioned here.

Response: We really appreciate the reviewer's thoughtful comments. We have included this point in the Introduction and Discussion parts.

l.94 Although the data in Fig 1b are convincing, it should be pointed out that only a very small fraction of pol eta (<1%) is bound to OGT

Response: Thanks for pointing out this. We have added the information in the revised version.

l.143-146 The PIP box at 707-708 is well established, whereas that at 443-444 is controversial. In addition F443-445 is quite close to T457, so the effect of mutating these residues may be more direct. Have the authors mutated the two PIP boxes separately? If not, these data should probably be omitted.

Response: We really appreciate the reviewer's thoughtful suggestions. We did not mutate the two PIP boxes separately. Based on the reviewer's suggestion, these data has been omitted in the

revised version.

l.147. The increase after UV does not look that great. Some quantitation and indication of reproducibility should be indicated. Also in view of the hypotheses proposed in the paper, it is important to see a time-course of the GlcNAcylation after UV.

Response: We really appreciate the reviewer's thoughtful suggestions. We have quantified the data and indicated the reproducibility in the revised figure legend. We have also included a time-course of the GlcNAcylation of Pol η after UV (Fig. 2a, bottom panel) to show that it changed dynamically.

l.174-175 FACS analysis should be provided to show whether the increased foci formation is the result of a higher proportion of cells in S phase.

Response: We have performed a FACS analysis to show that the cells expressing WT or T457A Pol η had comparable cell cycle profiles at 24 h post-UV or without UV treatment (Supplementary Fig. 2b), excluding the possibility that the increased Pol η foci at 24 h post-UV is the result of a higher proportion of cells in S phase.

l.198-199. In what cells were these mutagenesis experiments carried out? XP30RO?

Response: These mutagenesis experiments were carried out in GFP-, WT or T457A-complemented XP30RO cells. We have included this in the revised version.

Fig 3f should be plotted as log (survival) against dose. Was caffeine included in the post-irradiation incubation?

Response: The fig. 3f has been plotted as log (survival) against dose following the reviewer's suggestion. Yes, caffeine (0.4 mM) was included in the post-irradiation incubation media.

Fig 4 – a few questions. In a, what is the upper band in the siDDB1 lane. In b, why are the FLAG-pol eta bands so weak, compared to other panels, eg c, d, e? In panels c,d please explain exactly how the ratio was calculated. Why are panels e and f on the right and g and h on the left. This is confusing.

Response: In Fig. 4a, the upper band in the siDDB1 lane is a nonspecific band. We have repeated the experiment and replaced the original Fig.4a.

In Fig. 4b, we only showed a short-exposure of Pol η blot in the original submission. A longer exposure of Pol η blot has been included in the revised Fig. 4b.

In Fig. 4c, d, we apologize for not explaining it clearly. The ratio represented the relative polyubiquitination level of Pol η in each sample, with the polyubiquitination level of siNC-WT set to 1 (100%). Namely, the polyubiquitination levels of all other samples were normalized to that in siNC-WT. The grey densities of the polyubiquitination signals and those of unmodified Pol η were determined by Photoshop software (Adobe Systems Incorporated, USA). We have included this

information in the "Method" part.

We have rearranged the panels e, f, g and h in Fig.4 in the revised version for a clear presentation.

l.277-278. UV survival data for K462R should also be provided.

Response: We have established XP30RO cells stably expressing GFP, WT, T457A, K462R or T457A/K462R GFP-Pol η and performed UV survival assay. The result showed that K462R or T457A/K462R Pol η -complemented XP30RO cells exhibited similar level of UV hypersensitivity as T457A Pol η -complemented XP30RO cells (Supplementary Fig. 5f).

l.294, Fig 6c. Did the transfectants express wt and mutant pol eta at the same level?

Response: Yes, we used the XP30RO cells stably expressing either WT or T457A mutant for the experiment. A similar expression level in WT and mutant Pol η was demonstrated and this data was included in Fig. 2d.

l.314 This should be reworded to indicate substantial rescue in T457A cells but not up to wt levels.

Response: We appreciate the reviewer's suggestion. We have reworded the sentence as below: But after exposure to cisplatin, the median fiber length in T457A Pol η -expressing cells (8.62 μ m) was substantially shorter than that in WT Pol η -expressing cells (10.65 μ m) ($p=0.0036$), although it was longer than that in GFP-complemented cells (5.90 μ m) ($p=0.505$).

l. 332 Please delete "for the first time". Unnecessary and irritating.

Response: Thanks for pointing out this. This description has been deleted in the revised version.

l.792, fig 1f. Indicate meaning of SE and LE.

Response: We are sorry for not presenting them clearly in our initial submission. We have indicated SE and LE in the Figure legend (original Fig. 1f, now Fig. 1h). Namely: SE represents short exposure, LE represents long exposure.

Reviewer #3 (Remarks to the Author):

DNA Pol η plays a key role in Translesion DNA Synthesis (TLS). With an affinity purification approach, Ma et al. identified OGT as a functional partner of DNA Pol η . The authors further characterized that Pol η is O-GlcNAcylated at T457 via the interaction with OGT. This novel posttranslational modification suppressed the CRL4CDT2-dependent ubiquitination at K462, thus impaired TLS. Overall, the manuscript may reveal a novel molecular mechanism underlying TLS. However, compelling evidence is needed to justify the major conclusions in this manuscript, and current form is too preliminary to warrant publication.

Major points:

1: Little evidence indicates that O-GlcNAcylation is involved in TLS. The author cited studies from other groups to show the biological function of O-GlcNAcylation in DNA damage repair. However, none of these studies directly address the significance of O-GlcNAcylation in TLS. The authors need to establish the function of OGT or OGA in TLS before characterizing the possible O-GlcNAcylation sites in Polη.

Response: We totally agree with this reviewer that there are no reports in the literature to demonstrate that O-GlcNAcylation is involved in TLS although it was shown to participate in DNA damage repair. In this revised version, we have included additional experiment results that strongly support that OGT regulates TLS polymerase Polη function after UV irradiation: (1) We treated U2OS cells with UVC (15 J/m²) and collected the chromatin fractions at different time points. We found that the chromatin binding of OGT was enhanced after UV treatment (Fig. 1c). (2) Polη is an important TLS polymerase involving in UV-induced DNA damage tolerance. We showed that knockdown of OGT did not affect UV-induced Polη focus formation at 8 h, but caused a significantly higher extent of Polη foci at 24 h after UV irradiation compared with that of negative control (Fig. 1d).

2, In the manuscript, the authors used overexpression approach to examine the O-GlcNAcylation of Polη. However, the authors need to provide the evidence of endogenous O-GlcNAcylation on T457 of Polη.

Response: Thanks for this thoughtful suggestion. We would like to point out that the endogenous O-GlcNAcylation on T457 of Polη is hard to be determined mainly due to the extremely labile nature of O-GlcNAc and lack of suitable commercial Polη antibodies for endogenous Polη immunoprecipitation experiments. We have tried several Polη antibodies that had been used for immunoblotting in previous reported studies, including commercial anti-Polη antibodies from Santa Cruz (B-7, sc-17770) (Buisson R, et al. 2014. Cell Reports. 6: 553) and (H-300, sc-5592, discontinued) (Akagi J, et al. 2009 DNA Repair. 8: 585), Abcam (ab17725, currently out of stock for more than 1 year) (Bienko M, et al. 2010. Mol Cell. 37:396), Bethyl (A301-231A) (Day TA, et al. 2010. J Cell Biol. 191:953), and lab-prepared anti-Polη from Drs. Alan Lehmann and Simone Sabbioneda (Kannouche P, et al. 2004. Mol Cell. 14:491; Bienko M, et al. 2010. Mol Cell. 37:396). Unfortunately, none of these antibodies could immunoprecipitate enough endogenous Polη for an mass spectrometry assay.

In addition, although protein O-GlcNAcylation was discovered three decades ago, the extremely labile nature of O-GlcNAc makes it difficult to determine modification sites by common mass spectrometric approaches (Griffin ME, et al. 2016. Mol BioSyst. 12:1756-1759; Ma J & Hart GW. 2014. Clinical Proteomics. 11:8) or develop site-specific O-GlcNAc antibodies for target proteins (Ma J & Hart GW. 2014. Clinical Proteomics. 11:8). The currently common method to confirm the O-GlcNAc sites in target protein is to express WT and mutant constructs in cells, then examine whether mutation of the potential site(s) abrogates protein O-GlcNAcylation through immunoblotting with Pan O-GlcNAc antibodies. Using this approach, our studies have demonstrated the O-GlcNAcylation on T457 of Polη.

3. If OGT directly interacts with Pol η , it is essential for the authors to map the interaction regions, and perform the structure-function analysis.

Response: We appreciate the reviewer's suggestion. We have purified GST-OGT and incubated it with a panel of Flag-tagged truncated Pol η in GST pull down assays. We found that OGT interacts with multiple fragments, including Pol η (1-353), Pol η (350-590), Pol η (401-713). Thus, the structure-function analysis is hard to be performed.

4. The authors use the T457A mutation to claim the loss of O-GlcNAcylation, which is not accurate. As O-GlcNAcylation can compete with phosphorylation in many biological processes, this Thr residue may also be phosphorylated. The phenotype of the T457A mutation may result from the loss of phosphorylation.

Response: We appreciate the reviewer's thoughtful comments. To exclude the possibility that the phenotype of the T457A mutation may result from the loss of phosphorylation, we have constructed two phospho-mimetic mutants by replacing Thr457 with Asp (T457D) or Glu (T457E). We found that analogous to T457A mutant, T457D and T457E Pol η displayed a significantly higher extent of Pol η foci at 24 h after UV treatment (Supplementary Fig. 2c). Moreover, by using Mass-spec to identify potential phosphorylation sites on Pol η after UV treatment, T457 could not be identified, while the previous reported phosphorylation site S601 could be easily identified (Data not shown). These results indicated that the phenotype of the T457A mutation does not result from the loss of phosphorylation at T457.

5. The T457A mutation may simply abolish the consensus motif for CRL4^{CDT2}-dependent ubiquitination. The phenotype may not even result from the loss of O-GlcNAcylation or

phosphorylation per se. I strongly suggest the authors to use in vitro biochemistry assays to justify the major conclusions. A simple point mutation in the cell context may generate a lot of artifacts.

Response: We appreciate the reviewer's thoughtful comments. We have purified His-tagged Pol η from *Escherichia coli* Transetta (DE3) cells and CRL4^{CDT2} from 293T cells, and performed *in vitro* ubiquitination (Supplementary Fig. 4b-e). The result showed that WT and T457A Pol η exhibited similar extent of ubiquitination by CRL4^{CDT2} (Supplementary Fig. 4e), excluding the possibility that the T457A mutation may simply abolish the consensus motif for CRL4^{CDT2}-dependent ubiquitination.

We have also performed *in vitro* biochemistry assays based on the reviewer's suggestion. We transformed constructs expressing GST-OGT and His-tagged Pol η into *Escherichia coli* and purified His-Pol η followed by immunoblotting using the anti-O-GlcNAc antibody as described previously (Guo B., et al. 2014. Nat Cell Biol. 16:1215-26. doi: 10.1038/ncb3066). The result exhibited that co-expression of OGT resulted in an obvious O-GlcNAc-modification of WT Pol η , whereas T457A mutation significantly reduced Pol η O-GlcNAcylation (Supplementary Fig. 4b, c). We further used the purified proteins as substrates to do *in vitro* ubiquitination. We found that, after co-transformation with OGT, WT but not T457A manifested a remarkably increased ubiquitination compared to their controls without OGT co-transformation (Supplementary Fig. 4e).

More importantly, we also incubated the purified WT Pol η under the presence or absence of OGT co-transformation (Supplementary Fig. 4b, c) with cell lysates expressing Myc-DDB1 and Flag-CDT2 for *in vitro* pull down assay. We found that under the presence of GST-OGT, which promotes Pol η O-GlcNAcylation (Supplementary Fig. 4c), the association between Pol η and DDB1 or CDT2 was significantly enhanced, suggesting that Pol η O-GlcNAcylation promotes its interaction with CRL4^{CDT2} complex (Supplementary Fig. 4h).

These results properly justified our conclusion that Pol η O-GlcNAcylation at T457 facilitates its ubiquitination.

Reviewers' comments:

Reviewer #1 (Remarks to the Author):

The Authors addressed all my concerns and the new data provided has improved significantly the paper. For these reasons I fully endorse its publication on Nature Communications.

Reviewer #2 (Remarks to the Author):

The authors have addressed my comments satisfactorily.

Reviewer #3 (Remarks to the Author):

The authors have tried to address the major points with additional experiments. However, the results are not satisfactory.

1. In the revised manuscript, the authors showed that loss of OGT increased the foci of Pol η at 24 h after UV treatment (Fig. 1d). However, at that time point, O-GlcNAcylation of Pol η was reduced close to the level of untreated control (Fig. 2a), indicating that OGT might regulate the foci of Pol η via other mechanism. Furthermore, it is still unclear if OGT is required for the ubiquitination of Pol η , UV-induced mutation rate, and cell viability following UV treatment.

2. The authors seem to have difficulty to detect the O-GlcNAcylation on endogenous Pol η . However, it is essential to show that endogenous Pol η is able to be O-GlcNAcylated. Otherwise, the modification could be an artifact due to protein overexpression.

3. The authors need to use co-IP assays to identify the interactions regions on both OGT and Pol η . The essential binding regions can be identified using internal deletion mutants.

4. As the T457A mutation may abolish the tertiary structure or other modifications on T457 or adjacent sites, I suggested the authors to perform in vitro biochemistry assays to validate the results. However, the results shown in Supplemental Fig. 4e are ambiguous. It is unclear if the shifted signals represent ubiquitinated Pol η . Two additional bands can even be visualized, which may not represent ub Pol η .

Reviewer #3 (Remarks to the Author):

The authors have tried to address the major points with additional experiments. However, the results are not satisfactory.

1. In the revised manuscript, the authors showed that loss of OGT increased the foci of Polη at 24 h after UV treatment (Fig. 1d). However, at that time point, O-GlcNAcylation of Polη was reduced close to the level of untreated control (Fig. 2a), indicating that OGT might regulate the foci of Polη via other mechanism. Furthermore, it is still unclear if OGT is required for the ubiquitination of Polη, UV-induced mutation rate, and cell viability following UV treatment.

Response: We really appreciate the reviewer's thoughtful comments. Usually, UV-induced Polη foci reach the peak around 8 h and decline over time from about 12 h post-UV. We noticed that the level of O-GlcNAcylation of Polη was higher during 8-16 h and was reduced to about 1.77 fold at 24 h post-UV relative to that in untreated control (Fig. 2a). The dynamic status of O-GlcNAcylation of Polη is consistent with the model we proposed that O-GlcNAcylation of Polη occurs prior to Polη poly-ubiquitination and thereby removal by p97.

Furthermore, we determined the effects of OGT depletion on the ubiquitination of Polη, UV-induced mutation rate, and cell viability following UV treatment. We found that knockdown of OGT in 293T cells significantly abrogated Polη polyubiquitination (Supplementary Fig. 5a) and promoted UV light-induced mutagenesis (Supplementary Fig. 5b). Additionally, depletion of OGT led to an increased UV sensitivity in both U2OS and MRC5 cells (Supplementary Fig. 5c, d). These data clearly indicated that OGT is required for the ubiquitination of Polη, UV-induced mutation rate, and cell viability following UV treatment (revised supplementary Fig. 5).

2. The authors seem to have difficulty to detect the O-GlcNAcylation on endogenous Polη. However, it is essential to show that endogenous Polη is able to be O-GlcNAcylated. Otherwise, the modification could be an artifact due to protein overexpression.

Response: As mentioned in our previous response letter, examination of endogenous level of Polη O-GlcNAc is technically infeasible due to lack of suitable Polη antibodies for endogenous Polη immunoprecipitation experiments. To further address this comment, we have employed an alternative approach by taking advantage of an inducible XP30RO-Polη cell line (Han J, et al. 2014. J Cell Biol. 811-827) in which SFB (streptavidin-Flag-S protein)-tagged Polη was induced by Doxycycline (0.01 μg/ml) at a similar level as endogenous Polη expression. The SFB-Polη was then immunoprecipitated and immunoblotted with anti-O-GlcNAc antibody. The result indicated that Polη when expressed at the endogenous level was also O-GlcNAcylated (Supplementary Fig. 1a), excluding the possibility that Polη O-GlcNAcylation might be caused by its overexpression.

3. *The authors need to use co-IP assays to identify the interactions regions on both OGT and Polη. The essential binding regions can be identified using internal deletion mutants.*

Response: Based on the reviewer's suggestion, we mapped the regions within Polη responsible for its interaction with OGT by using Flag-tagged WT Polη and a series of Polη deletion mutants (Supplementary Fig. 2a). Coimmunoprecipitation experiments revealed that the N-terminal fragment spanning the whole catalytic domain of Polη is required for its interaction with OGT (Supplementary Fig. 2b). Similarly, we also mapped the regions within OGT required for its association with Polη by using Myc-tagged WT OGT and a series of OGT deletion mutants (Supplementary Fig. 2c). Coimmunoprecipitation assays showed that the first two TPR regions in OGT are required for its association with Polη (Supplementary Fig. 2d).

4. *As the T457A mutation may abolish the tertiary structure or other modifications on T457 or adjacent sites, I suggested the authors to perform in vitro biochemistry assays to validate the results. However, the results shown in Supplemental Fig. 4e are ambiguous. It is unclear if the shifted signals represent ubiquitinated Polη. Two additional bands can even be visualized, which may not represent ub Polη.*

Response: We appreciate the reviewer's thoughtful comments. We repeated the *in vitro* ubiquitination assay and blotted the immunoprecipitated Polη with ubiquitin antibody instead of the Polη antibody (revised supplementary Fig. 6e). The result clearly showed that, without co-expression of OGT, WT and T457A Polη exhibited similar extent of ubiquitination, excluding the possibility that the T457A mutation may simply abolish the consensus motif for CRL4^{CDT2}-dependent ubiquitination. After co-expression of OGT, the ubiquitination level of WT but not T457A manifested a remarkably increase, indicating that Polη O-GlcNAcylation at T457 promotes CRL4^{CDT2}-mediated Polη polyubiquitination,

REVIEWERS' COMMENTS:

Reviewer #3 (Remarks to the Author):

The authors have addressed my questions. I recommend the manuscript for publication.